# Super-enhancer-based identification of a BATF3/ IL-2R−module reveals vulnerabilities in anaplastic large cell lymphoma

Huan-Chang Liang [1,2], Mariantonia Costanza [2,3,4], Nicole Prutsch[5], Mark W. Zimmerman [5], Elisabeth Gurnhofer[1], Ivonne A. Montes-Mojarro [2,6], Brian J. Abraham [7], Nina Prokoph [2,8], Stefan Stoiber [1,9], Simone Tangermann[10], Cosimo Lobello[2,11], Jan Oppelt [11], Ioannis Anagnostopoulos[12], Thomas Hielscher[13], Shahid Pervez[14], Wolfram Klapper [15], Francesca Zammarchi[16], Daniel-Adriano Silva [17,18,19], K. Christopher Garcia [20,21], David Baker [17,18,21], Martin Janz [3,4], Nikolai Schleussner[3,4], Falko Fend [2,6], Šárka Pospíšilová[2,11,22], Andrea Janiková[2,22], Jacqueline Wallwitz[23], Dagmar Stoiber[23], Ingrid Simonitsch-Klupp[1], Lorenzo Cerroni[24], Stefano Pileri[25], Laurence de Leval [26], David Sibon[27], Virginie Fataccioli[28], Philippe Gaulard[28], Chalid Assaf[29], Fabian Knörr[30], Christine Damm-Welk[2,30], Wilhelm Woessmann[2,30], Suzanne D. Turner [2,8,11], A. Thomas Look [5], Stephan Mathas [2,3,4,13,32✉], Lukas Kenner [1,2,9,10,31,32✉] & Olaf Merkel [1,2,32✉]

Anaplastic large cell lymphoma (ALCL), an aggressive CD30-positive T-cell lymphoma, comprises systemic anaplastic lymphoma kinase (ALK)-positive, and ALK-negative, primary cutaneous and breast implant-associated ALCL. Prognosis of some ALCL subgroups is still unsatisfactory, and already in second line effective treatment options are lacking. To identify genes defining ALCL cell state and dependencies, we here characterize super-enhancer regions by genome-wide H3K27ac ChIP-seq. In addition to known ALCL key regulators, the AP-1-member *BATF3* and *IL-2 receptor* (*IL2R*)-components are among the top hits. Specific and high-level IL2R expression in ALCL correlates with BATF3 expression. Confirming a regulatory link, IL-2R-expression decreases following *BATF3* knockout, and BATF3 is recruited to *IL2R* regulatory regions. Functionally, IL-2, IL-15 and Neo-2/15, a hyper-stable IL-2/IL-15 mimic, accelerate ALCL growth and activate STAT1, STAT5 and ERK1/2. In line, strong IL-2Rα-expression in ALCL patients is linked to more aggressive clinical presentation. Finally, an IL-2Rα-targeting antibody-drug conjugate efficiently kills ALCL cells in vitro and in vivo. Our results highlight the importance of the BATF3/IL-2R-module for ALCL biology and identify IL-2Rα-targeting as a promising treatment strategy for ALCL.

A full list of author affiliations appears at the end of the paper.

Systemic anaplastic large cell lymphoma (ALCL) is an aggressive CD30-positive, mature T-cell non-Hodgkin lymphoma (T-NHL), which can be divided into two entities according to the presence or absence of *Anaplastic Lymphoma Kinase* (*ALK*)-fusions, i.e., ALK-positive (ALK$^+$) and ALK-negative (ALK$^-$) ALCL[1,2]. Furthermore, breast implant-associated ALK$^-$ALCL (BIA-ALCL) and primary cutaneous ALCL (pcALCL) are acknowledged as (provisional) entities[2], in particular the latter with usually good prognosis[3]. Despite implementation of targeted treatment strategies such as CD30 antibody-drug conjugates (ADCs) into up-front treatment regimens[4], the prognosis of some particular systemic ALCL subtypes is unsatisfactory[5], and there is an unmet medical need for effective treatment options in the case of relapsed and refractory disease[6,7].

The AP-1 factors JUNB and BATF3, representing key components of the ALCL cell-specific transcriptional program[8–11], interact with IRF4[12] and are essential for growth, survival and TH17/ILC3-skewing of ALCL[12,13]. Remarkably, ALCL rarely express T-cell receptor (TCR)β and TCR-associated signaling molecules[14–16], suggesting that other pathways substitute for TCR-signaling. The maturation, proliferation and survival of T cells critically depends on IL-2[17]. IL-2 binds to IL-2 receptor (IL-2R)αβγ with high, and to the heterodimeric IL-2Rβγ with intermediate affinity[18]. IL-2R-expression has been linked to ALCL[19], and IL-2Rγ is epigenetically silenced in some ALCL, ALK$^+$ cell lines[20]. Whereas IL-2 is secreted by activated T-cells, IL-15 together with its receptor IL-15Rα is usually membrane-bound, and is presented to cells that express the IL-2Rβγ dimer. Thus, the receptors of IL-2 and IL-15 use both the IL-2Rβ and IL-2Rγ chains[18].

Super-enhancers (SEs) are genomic regions bound by large clusters of regulatory elements resulting in high-level gene expression of the respective genes, which are marked by extensive stretches of histone 3 lysine 27 acetylation (H3K27ac)[21,22] and help to define unique tumor dependencies and therapeutically relevant vulnerabilities[23].

Here, we perform a SE analysis by genome-wide chromatin immunoprecipitation sequencing (ChIP-seq) of H3K27ac in ALCL cells in order to characterize in an unbiased manner the transcriptional network and to identify key players that define ALCL pathogenesis. We find *BATF3* and *IL2R* components are among the top SE hits, prompting us to analyze the link between BATF3 and IL-2R components in ALCL cells, and to explore the biological function of the IL-2/IL-2R system in ALCL in detail.

## Results

**Characterization of the super-enhancers in ALCL.** To identify in an unbiased manner cellular dependencies and vulnerabilities which could ultimately be tools for ALCL therapy, we performed genome-wide H3K27ac ChIP-seq for SE detection in 8 ALCL cell lines (5 ALK$^+$, 3 ALK$^-$). By applying the ROSE algorithm[22,24], we defined SEs, characterized by extensive H3K27ac, in all 8 cell lines (Fig. 1a, b and Supplementary Fig. 1a, b). These were compared to published H3K27ac ChIP-seq datasets of CD3$^+$, CD8$^+$ and CD4$^+$ T-cell subsets from healthy donors as well as T-cell-derived Jurkat cell line (Supplementary Fig. 1b).

Globally, we defined 538 and 722 SEs in Karpas-299 (systemic ALCL, ALK$^+$) and Mac-1 (derived from cutaneous T-NHL later progressing to ALCL) cell lines (Fig. 1a, b), respectively. 225 genes associated with SEs were shared between both cell lines (Fig. 1c and Supplementary Table 1). The identified SEs were associated with transcription factors and genes considered key pathogenic factors in ALCL, including *CD30, IRF4, JUNB, STAT1* and *STAT3*[10,25,26]. Furthermore, in the analyzed ALCL cell lines, *BATF3*[12,27] was consistently found in a SE region with the highest

H3K27ac, as were the *IL2RA* and *IL2RB* loci (Fig. 1a, b and Supplementary Fig. 1a, b). Remarkably, localization of *BATF3* and *IL2RA* and/or *IL2RB* in SE regions was a common feature in ALCL cell lines but not in primary T-cell controls or the Jurkat cell line (Fig. 1a, b, c and Supplementary Fig. 1a, b), indicating an ALCL-specific property.

To test whether the SE pattern seen in ALCL cell lines is reflected in ALCL primary patient samples, we performed H3K27ac analysis using the cut-and-run ChIP-seq technique on 2 primary patient samples (Fig. 1d and Supplementary Fig. 1c). Overall, we found a similar SE pattern as in the ALCL cell lines. In both patients #54 and #208, *BATF3* and *IL2RB* were located in SE regions, and *IL2RA* was located in SE region in patient #54. Together, these data mirrored our findings in ALCL cell lines, showing the presence of IL2R components and BATF3 in SE regions in ALCL.

**BATF3-regulated genes and pathways in ALCL.** To identify BATF3-regulated genes, RNA-seq of Karpas-299 cells after CRISPR/Cas9-mediated *BATF3* deletion was performed. Among the most significantly downregulated genes in 3 different CRISPR/Cas9-mediated *BATF3*-deleted clones was *IL2RG* (Supplementary Fig. 2a), and IL-2- together with in ALCL well-described IL-10- and STAT3-signaling was detected as being deregulated[25,28] (Fig. 1e and Supplementary Fig. 2b). In order to distinguish direct from indirect BATF3-regulated genes, we performed genome-wide BATF3 ChIP-seq in Karpas-299 cells. 354 of 581 (61%) downregulated genes after *BATF3* CRISPR-inactivation showed BATF3-binding in their regulatory regions, indicating direct regulation by BATF3, whereas only 368 of 865 (43%) upregulated genes showed BATF3-binding peaks (Supplementary Table 2). Remarkably, BATF3 occupied H3K27ac-enriched SEs presenting at the *IL2RA* and *IL2RB* loci (Supplementary Fig. 2c). Furthermore, binding of BATF3 at its own locus suggests an autoregulatory loop, as often observed with SE transcription factors (Supplementary Fig. 2d). Together, the SE analyses and BATF3 target gene characterization suggested a central role for BATF3 and IL-2/IL-2R signaling in ALCL.

Next, we explored publicly available RNA-seq data from 23 ALCL patients[29] for genes positively correlated with *BATF3* (Fig. 1f). Among 21,000 assigned and expressed genes, *CD30* and *IL2RA* were within the top 50 hits positively correlating with *BATF3* (both $P < 0.001$). Also, *IL2RB* positively correlated with *BATF3* expression ($P = 0.027$; Fig. 1f). The same pattern was observed in an independent microarray dataset of 29 ALCL patients[30] (Supplementary Fig. 2e). In line, BATF3 and IL-2Rα expression positively correlated ($P = 0.0017$) by immunohistochemistry (IHC) of ALCL specimens using a tissue microarray (TMA) ($n = 63$; Fig. 1g, Supplementary Fig. 2f and Supplementary Tables 3 and 4).

**IL-2Rα/β/γ are highly expressed in ALCL but not other T-NHL.** To comprehensively examine expression of the IL-2R subunits, IL-2Rα (also called CD25), IL-2Rβ (CD122) and IL-2Rγ (CD132), in mature T-NHL, we performed immunostaining of TMA comprising angioimmunoblastic T-cell lymphoma (AITL, $n = 8$), peripheral T-cell lymphoma not otherwise specified (PTCL-NOS, $n = 23$), ALCL with ($n = 22$) or without ($n = 23$) *ALK*-translocation, and of non-neoplastic reactive lymph node controls ($n = 11$) as well as of pcALCL specimens ($n = 24$). Strong positive staining of all three IL-2R chains was seen in the majority of cases of all ALCL subtypes, with the strongest expression in ALCL, ALK$^+$, whereas no comparable IL-2R expression was observed in the other samples (Fig. 1h, Supplementary Fig. 3a and Supplementary Tables 3 and 4). The BATF3 expression analysis showed

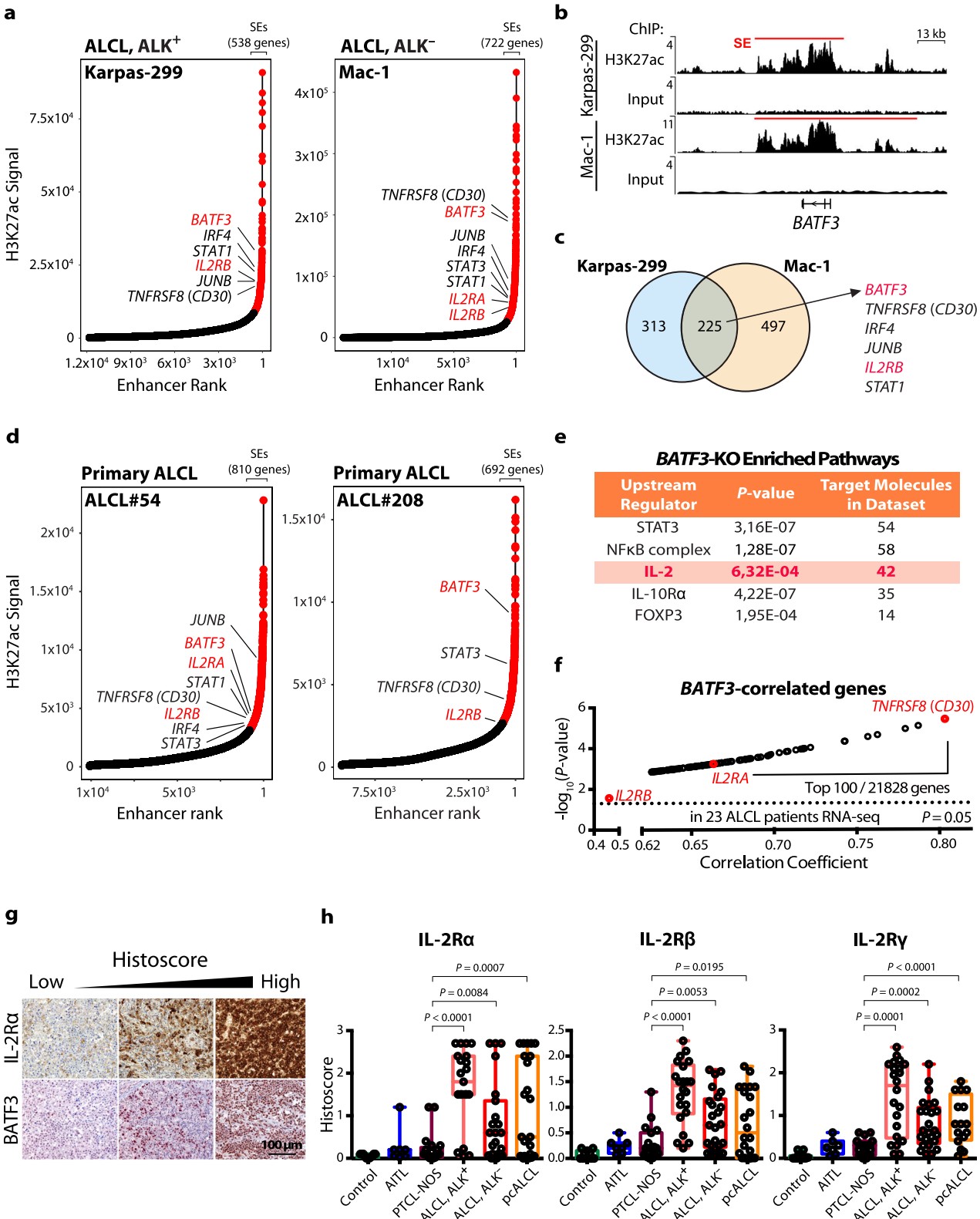

a similar pattern (Supplementary Fig. 3b and Supplementary Table 3), again suggesting a link between BATF3 and IL-2R expression in ALCL. We subsequently tested *IL2R* and *BATF3* expression in a broad cell line panel consisting of ALCL, ALK[+], ALCL, ALK[−], BIA-ALCL[31] and T-cell control cell lines. Again, expression of *IL2RA*, *IL2RB* and *BATF3* was restricted to ALCL cell lines (Fig. 2a and Supplementary Fig. 3c, d, e).

**BATF3 directly regulates IL-2Rα/β and CD30 in ALCL**. To assess if a direct link exists between BATF3 and IL-2R expression, we examined the mRNA levels of *IL2R* in *BATF3*-knockout ALCL cell lines. *BATF3* deletion led to significantly reduced mRNA and protein expression of all three IL-2R subunits in all tested ALCL cell lines, and of IL-2Rα and IL-2Rγ in BIA-ALCL cells (Fig. 2b and Supplementary Fig. 3f, g). Moreover, *BATF3* deletion led to

**Fig. 1 Genes encoding BATF3 and subunits of the IL-2R are highly acetylated at H3K27 and positively correlate. a** Enhancers were ranked based on increasing H3K27ac signals. Genes within SEs in Karpas-299 (ALK[+]) and Mac-1 (ALK[−]) ALCL cell lines are marked red. **b** H3K27ac ChIP-seq tracks at the *BATF3* locus with indicated SE regions. **c** Overlap of genes associated with SEs in Karpas-299 and Mac-1 cell lines. **d** Enhancers were ranked based on increasing H3K27ac signals. Genes within SEs in primary ALCL patient samples (#54 and #208) are marked in red. **e** Identification of upstream regulators among deregulated genes in CRISPR/Cas9-mediated *BATF3*-KO Karpas-299 cells by IPA 2020 analysis. **f** Analysis of *BATF3*-correlated genes in a previously published RNA-seq dataset of 23 ALCL patients (BioProject PRJNA255877, SRA identifier SRP044708). Pearson correlation; *P* value based on t-distribution. **g** Representative images of 63 patient samples measured of BATF3 and IL-2Rα IHC staining of ALCL, ALK[+] in paired FFPE tissues sections. **h** IHC quantification of IL-2Rα, IL-2Rβ and IL-2Rγ expression in mature T-NHL TMAs (reactive lymph node controls, *n* = 11; AITL, *n* = 8; PTCL-NOS, *n* = 23; ALCL, ALK[+], *n* = 22; ALCL, ALK[−], *n* = 23) and pcALCL (*n* = 24) specimens. *P* values were determined by two-tailed unpaired Student's *t* test. All box-whisker plots represent the median (central line), 25th–75th percentile (bounds of the box) and minimum–maximum (whiskers).

reduced growth in ALCL cell lines (Supplementary Fig. 3h). Protein and pY-protein levels of STAT1, STAT5 and ERK1/2 were not consistently affected by CRISPR/Cas9-mediated *BATF3* deletion (Supplementary Fig. 3i).

Both CD30 and BATF3 display specificity for ALCL[11], and JUNB is present on the AP-1 site within the *CD30* promoter[32,33]. Given our identification of SEs at *CD30*, *BATF3* and *IL2R* loci, and that CD30 and IL-2R are markers for activated lymphocytes, we tested whether BATF3 also regulates CD30. CD30 expression decreased following *BATF3* deletion with a stronger effect in ALCL, ALK[+] cells (Fig. 2c), and BATF3 was recruited to *CD30* regulatory regions as revealed by ChIP (Fig. 2d) and genome-wide ChIP-seq (Fig. 2e). To independently assess AP-1 recruitment to regulatory regions of *IL2R* members in ALCL, we identified AP-1/TRE sites in the promoter or enhancer regions of *IL2R* loci. The non-conserved 5'-TGAGTAA-3' motif in the *IL2RA* promoter and the conserved 5'-TGACTCA-3' AP-1/TRE site in the *IL2RB* enhancer showed strong ALCL-specific DNA-binding (Fig. 2f and Supplementary Fig. 3j). Recruitment of BATF3 to *IL2RA* and *IL2RB* regulatory regions was also observed by ChIP in ALCL cell lines (Fig. 2g), confirming our BATF3 ChIP-seq in Karpas-299 and Mac-1 ALCL cells (Supplementary Fig. 2c). Together, our data point towards a direct regulation of *IL2R* and *CD30* genes by BATF3/AP-1 in ALCL.

IL-2 could not be detected in tumor cells of 33 ALCL patients nor in ALCL cell lines (Supplementary Fig. 4a, b and Supplementary Table 3), suggesting other sources of IL-2 in ALCL. Macrophages and plasma cells that do express IL-2 have been described as being in proximity to ALCL cells[34], which we did also observe in a number of cases (Supplementary Fig. 4a). Furthermore, elevated IL-2-levels were previously found in the serum of a subgroup of ALCL patients but none was detected upon remission or in B-NHL controls[35]. These data suggest that IL-2 might be present at least in a subset of ALCL patients and to act on the lymphoma cells.

**IL-2/-15 promote ALCL growth and STAT1/5 and ERK signaling.** Thus, to functionally investigate IL-2/IL-2R signaling in ALCL, we stimulated a panel of 7 ALCL cell lines with recombinant human (rh) IL-2. With the exception of SUP-M2, lacking the essential IL-2Rγ chain for IL-2 signaling, all ALCL cells showed significantly increased metabolic activity reflecting cell growth after rhIL-2 stimulation (Fig. 3a). Jurkat cells, lacking IL-2Rα and IL-2Rβ, were used as an additional control and did not respond to rhIL-2. Also, no growth stimulation of rhIL-2 was seen in ALCL cell lines following *BATF3* knockout (Fig. 3b). We also employed Neo-2/15, an IL-2-mimic that binds to heterodimeric IL-2Rβγ and is unrelated in amino acid sequence and topology to IL-2[36], which showed the same growth-promoting effect as rhIL-2 in ALCL cells with, but not without, BATF3 expression (Supplementary Fig. 4c, d). Stimulation with these cytokines led to increased phosphorylation levels of STAT1, STAT5 and ERK1/2, but not STAT3, in ALCL cell lines except for SUP-M2 (IL-2Rγ-negative) and Jurkat

(IL-2Rα- and IL-2Rβ-negative; Fig. 3c and Supplementary Fig. 4e, f). In limiting dilution assays on Karpas-299 and Mac-1 cells, rhIL-2 supported cell growth at low cell densities (Fig. 3d). The IL-2Rα-blocking antibody basiliximab attenuated rhIL-2-mediated growth-promoting effect (Fig. 3e) and activation of STAT5 and ERK1/2 (Supplementary Fig. 4g), confirming the specificity of observed effects.

IL-2R family members IL-2Rβ and IL-2Rγ together with IL-15Rα form the trimeric receptor complex for IL-15[17,18]. *IL15RA* was expressed in all ALCL cell lines, whereas *IL15* only in some (Supplementary Fig. 5a, b). Positive IHC staining for IL-15Rα (Supplementary Fig. 5c) and IL-15 (Fig. 3f and Supplementary Table 5) was also seen in primary ALCL samples. Expression of IL-15 and IL-15Rα together with high-level IL-2R suggests IL-15R-complex formation. In line, rhIL-15 stimulation enhanced cell growth in the majority of tested ALCL cell lines (Fig. 3g) via activation of the same downstream targets as IL-2 signaling, STAT1, STAT5 and ERK1/2 (Fig. 3h and Supplementary Fig. 5d, e). These results further support the biological significance of high-level IL-2R expression and interacting signaling components for ALCL.

**IL-2Rα level reflects clinical features of ALCL patients.** We next determined IL-2Rα expression in 88 pediatric ALCL, ALK[+] patients of the NHL-Berlin-Frankfurt-Münster (BFM) study group (German patients included in the ALCL99 trial[37] or the NHL-BFM trials[38]). Patients with high IL-2Rα expression of the tumor (staining intensity (SI) = 3) had lower five-year overall survival (OS) (Fig. 4a) compared to those with low expression (SI = 0–2). Furthermore, we analyzed an independent cohort of 44 adult ALCL, ALK[−] patients from the French, Swiss and Czech lymphoma study groups (Fig. 4b, Supplementary Fig. 6a and Supplementary Table 6). Estimates for event-free survival (EFS) and OS at 3 years for all 44 patients by IL-2Rα staining intensity showed a possible difference in EFS, not reaching significance (EFS 32% [95%-CI, 14–51%] for strong IL-2Rα positivity (SI = 3) vs. 50% [95%-CI, 28–68%], *P* = 0.11) (Supplementary Fig. 6a). First-line treatment for the patients was comparable in terms of induction chemotherapy, but differed in consolidation strategy, as 10 of 44 patients received an autologous or allogeneic blood stem cell transplantation (SCT) first-line (see Supplementary Table 6). Therefore, we performed a subgroup analysis for patients not receiving SCT as first-line consolidation. In this analysis, EFS was significantly lower in patients with tumors showing strong (SI = 3) IL-2Rα staining (24% [95%-CI, 7–45%] vs. 59% [95%-CI, 33–78%], *P* = 0.025) (Fig. 4b, left). A possible difference was observed in OS (*P* = 0.056), not reaching significance (Fig. 4b, right).

**Targeting IL-2Rα induces ALCL death in vitro and in vivo.** Given the biological role of IL-2/IL-2R in ALCL, the recent development of IL-2Rα-targeting strategies[39,40], and the medical

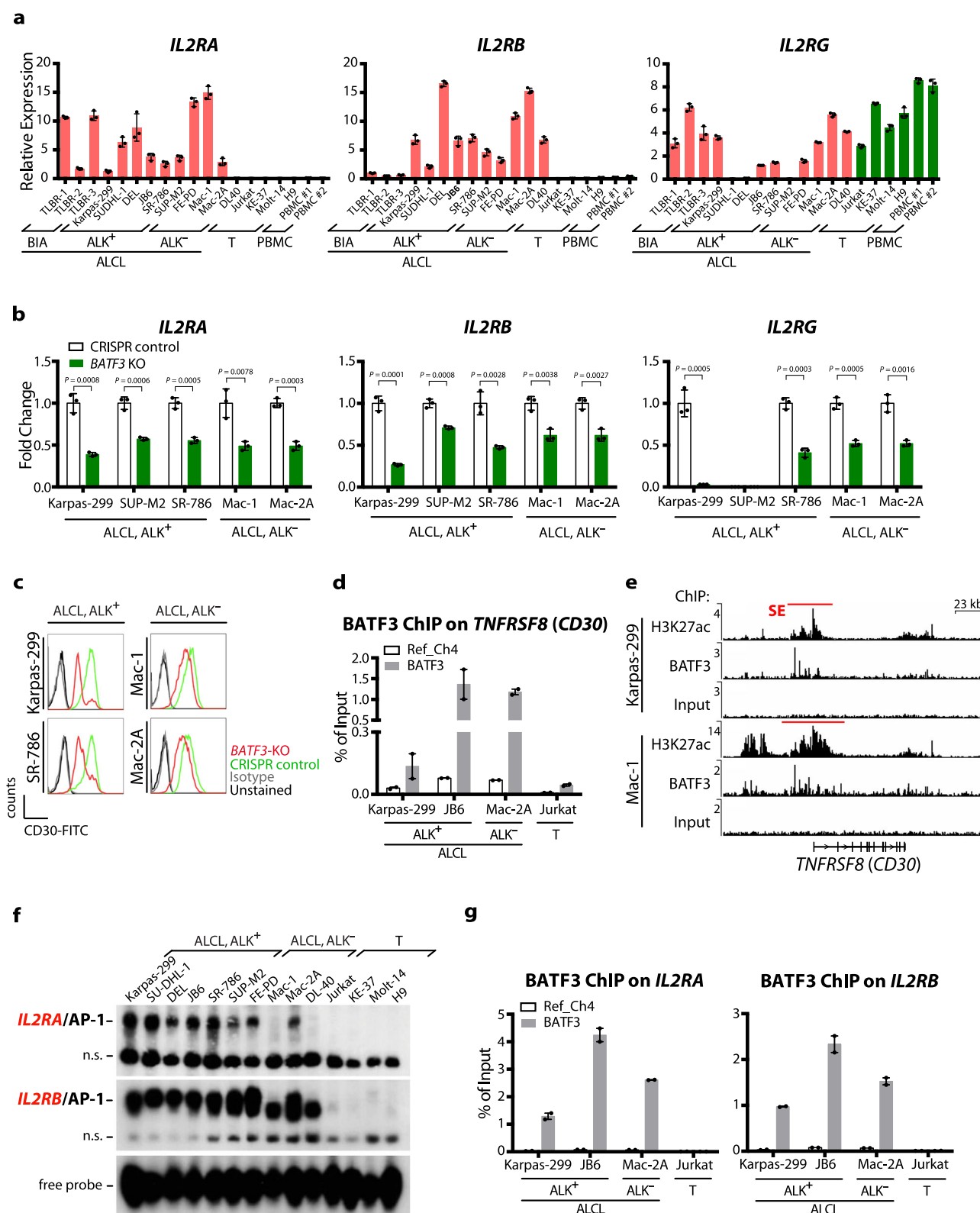

need for new treatment approaches for subgroups of ALCL, we explored IL-2Rα as therapeutic target in ALCL. To this end, we used ADCT-301 (camidanlumab tesirine), a recently developed ADC composed of an IL-2Rα-targeting monoclonal antibody coupled to a highly-potent pyrrolobenzodiazepine (PDB)-dimer cytotoxic warhead[39]. Cells were treated with ADCT-301 or the isotype control ADC B12-SG3249, recognizing HIV-gp120,

conjugated to the same linker-warhead as in ADCT-301. At doses of 156 ng/ml and lower, ADCT-301 induced cell death of all IL-2Rα-positive ALCL cell lines while it did not affect viability of IL-2Rα-negative Jurkat and KE-37 cells (Fig. 4c). B12-SG3249 did not affect cell viability. The LD50 of ADCT-301-induced cell death was remarkably low (between 0.8 and 2.3 ng/ml) for ALCL cell lines, but >156 ng/ml in the IL-2Rα-negative cell lines

**Fig. 2 BATF3 regulates expression of IL2R subunits and CD30 in ALCL. a** *IL2RA, IL2RB* and *IL2RG* mRNA expression in cell line panel consisting of 13 ALCL cell lines (6 ALCL, ALK⁺, 4 ALCL, ALK⁻ and 3 BIA-ALCL), 4 T-cell leukemia-derived control cell lines (T) and PBMCs, or in **b** CRISPR/Cas9-mediated *BATF3*-KO and CRISPR control ALCL cells, using quantitative RT-PCR. Data are indicated as means ± SD of biological triplicates. *P* values were determined by two-tailed unpaired Student's *t* test. **c** Cell surface expression of CD30 in *BATF3*-KO (red) or CRISPR control (green) ALCL cells, isotype control (gray) and unstained cells (black). **d** BATF3 ChIP of the *CD30* regulatory region as compared to a control region (Ref_Ch4). Combined data of two biological replicates are shown as mean ± SEM (see Supplementary Table 8 for ChIP primer sequences). **e** ChIP-seq tracks at the *CD30* locus for BATF3 and H3K27ac in ALCL cell lines with indicated SE regions. **f** Electrophoretic mobility shift assay (EMSA) of AP-1 TRE complexes on the *IL2RA* promoter or *IL2RB* enhancer region (see Supplementary Table 8 for probe sequences) in 10 ALCL cell lines and 4 T-cell leukemia-derived control cell lines. The positions of AP-1 complexes, nonspecific bands (n.s.) and free probes are indicated. Blots shown are representative of three independent experiments with similar results. **g** BATF3 ChIP on *IL2RA* or *IL2RB* regulatory regions as compared to a control region (Ref_Ch4). Combined data of two biological replicates are shown as mean ± SEM (see Supplementary Table 8 for ChIP primer sequences).

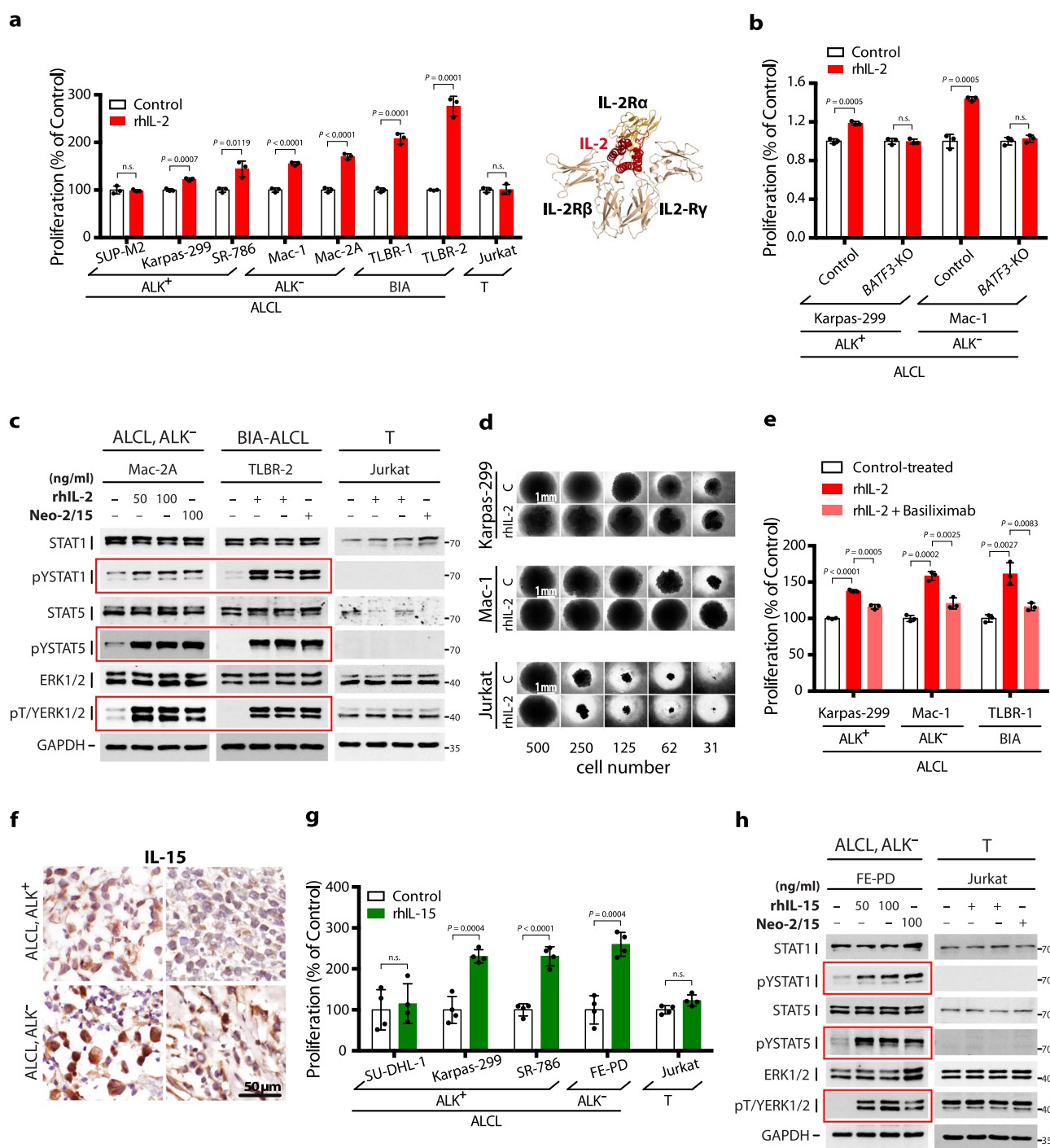

**Fig. 3 IL-2 and IL-15 promote ALCL proliferation and activate STAT1, STAT5 and ERK1/2. a** Indicated 7 ALCL cell lines were treated with rhIL-2 (100 ng/ml). Control cell lines were SUP-M2 (ALK+, IL-2Rγ-negative) and Jurkat (T-cell leukemia-derived, IL-2Rα- and IL-2Rβ-negative). Resazurine assay was used to measure metabolic activity as a marker of cell proliferation. Data are means ± SD of biological triplicates given as a percentage of the untreated control. *P* values were determined by two-tailed unpaired Student's *t* test; n.s. not significant. Structure of IL-2 bound to the trimeric IL-2R complex is shown (PDB ID: 2B5I). **b** In *BATF3*-KO cells, no IL-2-induced proliferation in Karpas-299 (ALK+) and Mac-1 (ALK−) could be seen compared to CRISPR control. Proliferation was determined by resazurine assay. Data are normalized to untreated control and shown as means ± SD of biological triplicates. *P* values were determined by two-tailed unpaired Student's *t* test. **c** Immunoblot analysis of total STAT1, STAT5 and ERK1/2 proteins and their phosphorylated (pY, pT/Y) forms in Mac-2A (ALK−), TLBR-2 (BIA) and Jurkat control cells stimulated with rhIL-2 (50 or 100 ng/ml), Neo-2/15 (100 ng/ml) or PBS (−) as control. GAPDH serves as loading control. Blots shown are representative of two independent experiments with similar results in different cell lines. **d** Limiting dilution of Karpas-299 (ALK+), Mac-1 (ALK−) and Jurkat control cells with or without rhIL-2 (100 ng/ml) in a 96-well plate containing RPMI1640 and 10% FCS. At day 0, the indicated number of cells were seeded and incubated for 2 weeks. Representative example of biological triplicates is shown. **e** Inhibition of IL-2-induced proliferation by the IL-2Rα-blocking antibody Basiliximab (1 μg/ml). Proliferation was determined by resazurine assay. Data are normalized to control-treated cells and shown as means ± SD of biological triplicates. *P* values were determined by two-tailed unpaired Student's *t* test. **f** Representative images of 9 patient samples measured of IL-15 IHC staining in fresh-frozen primary ALCL tissue. **g** Effect of IL-15 (50 ng/ml) on cell proliferation of indicated 4 ALCL and Jurkat cell lines. SU-DHL-1 (ALK+, IL-2Rγ-negative) and Jurkat (T-cell leukemia-derived, IL-15Rα- and IL-2Rβ-negative) serve as control cell lines. Proliferation was measured by [³H]-thymidine incorporation. Values represent means ± SD of biological triplicates as percentage of untreated control. *P* values were determined by two-tailed unpaired Student's *t* test; n.s. not significant. **h** Immunoblot analysis of STAT1, STAT5 and ERK1/2 proteins and their phosphorylated (pY, pT/Y) forms in FE-PD (ALK−) and Jurkat control cell lines stimulated with rhIL-15 (50 or 100 ng/ml), Neo-2/15 (100 ng/ml) or PBS (−) as control. GAPDH protein expression serves as control.

(Fig. 4c), indicating the high efficacy and specificity of ADCT-301 for ALCL. Annexin V/propidium-iodide (PI) staining and PARP1-cleavage (Supplementary Fig. 6b, c) in ADCT-301-treated cells indicated induction of apoptotic cell death. When we tested the effect of ADCT-301 or control ADC B12-SG3249 on STAT1 and STAT5 levels in ALCL cell lines, we found reduction of pYSTAT5 in all tested cell lines and reduction of pYSTAT1 in the FE-PD cell line (Supplementary Fig. 6d).

For determination of ADCT-301 efficacy in vivo, we subcutaneously xenotransplanted three ALCL, ALK− cell lines, Mac-2A, TLBR-1 and FE-PD, into NOD-*scid Il2rg^null* (NSG) mice and intravenously administered a single dose (0.5 mg/kg) of ADCT-301, control ADC B12-SG3249, or vehicle (PBS) when tumors reached 100–150 mm³. This single dose of ADCT-301 completely prevented growth of Mac-2A and TLBR-1 tumors and led to tumor growth retardation of rapidly growing FE-PD cells (Fig. 4d and Supplementary Fig. 6e). The control ADC did not elicit any anti-tumor activity. This remarkable tumor control of ADCT-301 confirmed the high efficacy of IL-2Rα-mediated targeting of ALCL cell growth.

## Discussion
ALCL is a heterogeneous disease comprised of systemic ALCL, ALK+ and ALK−, pcALCL and BIA-ALCL. Whereas the prognosis of adult ALCL, ALK+ and subgroups of ALCL, ALK− is usually good, relapses occur frequently in pediatric ALCL, ALK+, and the outcomes of patients with relapsed/refractory ALCL patients and advanced BIA-ALCL are rather poor, highlighting an unmet medical need for these patients[2,7].

Ideally, therapeutic targets are not only specific to the respective tumor cells, but, at the same time, are essential components of growth and survival pathways. Screening for SEs is a valuable tool for identification of such genes[21,22,24]. In our SE analyses, the identification of known ALCL-promoting genes, such as *JUNB*, *IRF4*, *STAT1*, *STAT3* and *CD30*[8,9,25,26,41], validated our approach and independently confirmed previous findings on the pathogenic role of JUNB, IRF4 and STAT transcription factors in ALCL. Of note, *BATF3*, *IRF4* and *STAT3* are present in the top-ten "preferentially essential genes" in ALCL, ALK+ cell lines identified by CRISPR screen analyses within the Achilles project (depmap.org; Supplementary Fig. 6f)[42]. Furthermore, *BATF*-family and *IRF4* genes are involved in chromosomal translocations in ALCL, supporting their biological relevance[29,43].

Remarkably, both the AP-1-family member BATF3 and IL-2R components were consistently found among the top SE-associated genes in ALCL cell lines and primary ALCL samples, which attracted our attention for detailed analyses given our previous findings on AP-1/BATF function in ALCL and the important role of IL-2/IL-2R for T lymphocyte growth and function[12,18]. We further extend the ALCL pathogenic concept on IL-2/IL-15/IL-2R-components and define the BATF3/IL-2R module as a central component for ALCL growth and survival (Fig. 4e). Furthermore, despite harboring *JAK2* or *STAT3* mutations[44], ALCL, ALK− cell lines strongly responded to IL-2, underscoring its growth-supportive role in ALCL. IL-2Rγ was described to be suppressed by NPM1-ALK[20]. However, according to our data (Fig. 1h), IL-2Rγ-silencing rarely occurs in ALCL. Regarding IL-2, its lack of expression in ALCL cells itself may be explained by their strong IL-10 expression, which can actively suppress IL-2 production in T cells[45]. However, cells in the ALCL tumor microenvironment can be assumed to produce IL-2[34,35], and in the majority of cases, ALCL cells produce IL-15, both accounting for IL-2R stimulation in primary ALCL.

Our data suggest that high-level IL-2Rα expression might be associated with a more aggressive clinical presentation in two independent ALCL cohorts, which has to be confirmed in larger ALCL cohorts. However, this finding reflects our functional studies and is in line with the correlation of high soluble IL-2Rα levels with more aggressive disease and lower EFS at 3 years in ALCL, ALK+ children cohorts[35]. As risk factor, IL-2Rα expression might be outcompeted by more intense treatment regimens with brentuximab-vedotin (BV)-containing regimens or SCT in our adult ALCL, ALK− cohort. Moreover, it is unlikely that *DUSP22* and *TP63* rearrangements, having been described in ALCL, ALK−[43,46], are confounding factors in our survival analysis based on the following findings. In the first clinical report, the 5-year OS of 21 patients with *DUSP22* rearrangement was 90%[7]. However, in a later work from the British Columbia Cancer Lymphoid Cancer database, the 5-year OS of 12 patients with *DUSP22* rearrangement was 40%[47]. Thus, the prognostic impact of *DUSP22* rearrangement is currently unclear, and it may be more related to prognostic factors such as IPI. In addition, in pcALCL, no impact of *DUSP22* rearrangement (present in about 30% of cases) on prognosis was observed[48]. Despite that *TP63* rearrangement has a confirmed poor prognosis in ALCL, ALK−, it is indeed very rare. We also revealed a BATF3-dependency of

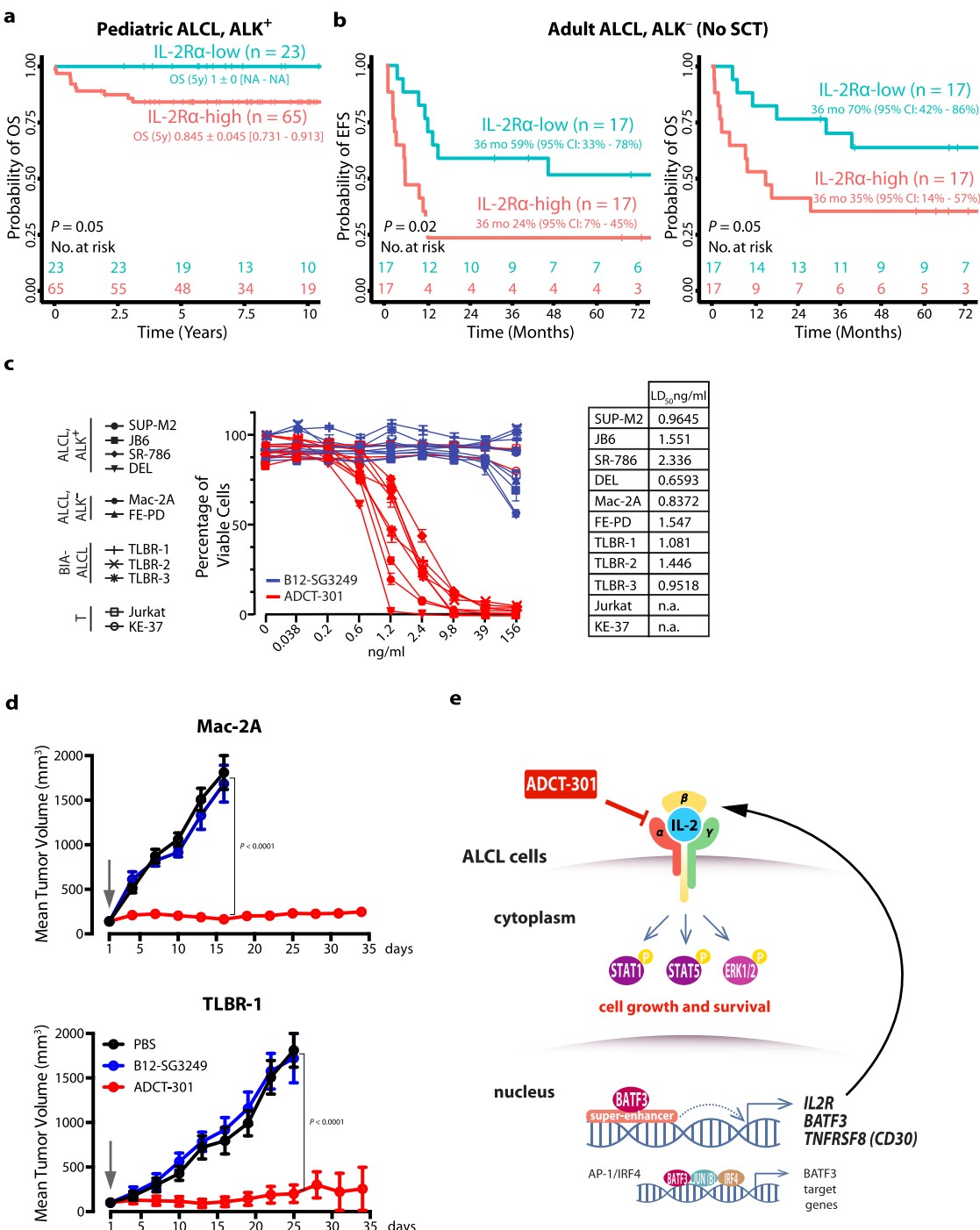

**Fig. 4 IL-2Rα expression in ALCL patients correlates with survival and represents potential therapeutic target. a** OS of pediatric ALCL, ALK+ patients (NHL-BFM cohort, $n = 88$), and **b** EFS and OS of adult ALCL, ALK− patients without SCT ($n = 34$) having low (staining intensity, SI = 0–2) or high (SI = 3) IL-2Rα expression. Kaplan–Meier curves of individual groups were compared using two-tailed log-rank statistics. **c** Indicated 9 ALCL cell lines and 2 T-cell leukemia-derived control cell lines were incubated for 96 h with different concentrations of PDB dimer linked antibodies targeting either IL-2Rα (ADCT-301) or, as a non-binding control, the glycoprotein gp120 of HIV (B12-SG3249) to determine the LD50. Means ± SD of biological triplicates are shown. **d** In vivo anti-tumor efficacy of ADCT-301 in murine xenograft models of Mac-2A (ALK−) or TLBR-1 (BIA) ALCL cell lines. For each cell line, mice were randomized into three groups to receive a single dose of either vehicle (PBS, $n = 4$), control ADC (B12-SG3249, 0.5 mg/kg, $n = 5$), or ADCT-301 (0.5 mg/kg, $n = 5$) intravenously at day 1 (indicated by an arrow). Tumor volumes were measured with caliper over time and are shown as means ± SEM. $P$ values were determined by two-tailed unpaired Student's $t$ test. **e** Schematic illustration of the proposed mechanism.

the ALCL hallmark-gene *CD30*. Whether IL-2R and CD30 are interconnected and IL-2R-signaling substitutes for the loss of TCR-signaling in ALCL[15,16] has to be determined in future studies.

Given the unsatisfactory treatment response of particular subgroups of systemic ALCL, ALK− patients, relapsed/refractory systemic ALCL patients or relapsed pediatric ALCL, ALK+ patients to standard chemotherapy, there is a medical need for

new effective treatment strategies[49]. With implementation of chemotherapy regimens containing BV, which proved to be the first effective targeted treatment for ALCL in many years[4], in frontline ALCL therapy, effective treatment options remain lacking in the second line. For ALCL, ALK+ patients, pharmacological ALK-inhibition might be a treatment option in this setting[50,51]. IL-2Rα has been investigated as immunotherapeutic tool for other malignancies[52,53]. ADCT-301, which has entered clinical trials (e.g., ClinicalTrials.gov Identifier: NCT02432235 and NCT04052997), killed ALCL cells with high efficacy[39]. Notably, ADCT-301-coupled PBD dimers, given their unique mode of action, are likely to overcome resistance to conventional chemotherapy[54]. Interestingly, ADCT-301 treatment in preclinical models and in patients with advanced solid tumors or lymphomas resulted in transient reduction of $T_{reg}$ cells, while CD4+ and CD8+ T cells were unaffected, which might enhance ADCT-301-mediated anti-tumor activity[55–57]. Together, by our unbiased approach we identify the BATF3/IL-2R-module as vulnerability in ALCL, and our data strongly argue for exploration of IL-2Rα-targeting of ALCL in clinical trials.

## Methods

**Cell lines and cell culture.** ALCL (Karpas-299, SU-DHL-1, DEL, JB6, SR-786 and SUP-M2, all ALK+; FE-PD, Mac-1, Mac-2A and DL-40, all ALK−), T-cell leukemia-derived (Jurkat, KE-37, Molt-14 and H9) and HEK293T cell lines were cultured as described[9,28]. If not stated otherwise, ALCL cell lines were obtained from the ATCC cell line repository. Non-ALCL T-cell leukemia-derived cell lines were received from the DSMZ collection (Braunschweig, Germany). Cell lines are regularly tested for mycoplasma contamination, and cell line identity has been verified by STR DNA fingerprinting (R. Siebert, Ulm, Germany). BIA-ALCL (TLBR-1, TLBR-2 and TLBR-3) cell lines were kindly provided by Alan Epstein (California, USA) and cultured in RPMI 1640 supplemented with 10% FBS, 100 IU/ml penicillin, 50 mg/ml streptomycin sulfate and 330 ng/ml in-house synthesized IL-2-Fc. FE-PD cell line was kindly provided by Annarosa del Mistro as a gift (Veneto Onocology Insitute Padua, Italy). Mac-1 and Mac-2A cell lines were kindly provided by Marshall E. Kadin as a gift (Harvard Medical School, Boston, MA). DL-40 was kindly provided by Masanori Daibata (Kochi, Japan).

Cell numbers were determined using a CASY Cell Counter. For cytokine stimulation, monoclonal antibody and ADC treatment, rhIL-2 (50 or 100 ng/ml; 1081-IL, R&D Systems), rhIL-15 (50 or 100 ng/ml; 247-ILB, R&D Systems), Neo-2/15[36] (100 nglml), basiliximab (1 µg/ml; ab00188, Absolute Antibody) and ADCT-301 (in vitro: 156 ng/ml, in vivo: 0.5 mg/kg; camidanlumab tesirine, ADC Therapeutics) were applied as indicated.

**Human samples.** All human samples were obtained with informed written consent and in accordance with the Declaration of Helsinki reviewed by the local ethic boards (Medical University of Vienna 1221/2019; University Hospital Brno 4-306/13/1). FFPE tissues from systemic ALCL, ALK− cases collected through TENOMIC, a transnational research consortium on T-cell lymphomas involving several centers in France, Belgium and Switzerland, were used for immunohistochemical validation. The TENOMIC database is approved by the ethical committee "CPP Ile-de-France IX 08-009".

**Genome-wide occupancy analysis.** ChIP-seq was performed as described[58,59]. Primary antibodies used for analysis are listed in Supplementary Table 7. In short, for each ChIP, 5 µg of antibody was conjugated to 2 mg of M-270 Epoxy Dynabeads (Thermo Fisher) and added to sonicated nuclear extract. Library construction was performed using the Swift 2S Turbo DNA library kit as per the manufacturer's instructions (Swift Biosciences). Illumina sequencing was performed using the NextSeq500 platform with single-end 75 bp reads.

**ChIP-seq processing and display.** Reads were aligned to the human genome (hg19) using bowtie with parameters –k 2 –m 2 –best and –l set to the read length. For visualization, WIG files were created from aligned ChIP-seq read positions using MACS with parameters –w –S –space=50 –nomodel –shiftsize=200 to artificially extend reads to be 200 bp and to calculate their density in 50 bp bins. Read counts in 50 bp bins were then normalized to the millions of mapped reads, giving reads per million (rpm) values. WIG and TDF files were visualized in the IGV 2.4.6 genome browser.

**ChIP-seq enriched regions.** Regions enriched in ChIP-seq signal were identified using MACS with corresponding control and parameters –keep-dup=auto and –p 1e−9. The distribution of peaks into other genomic features was determined by

overlapping peaks with SEs, typical enhancers, promoters, and gene bodies, in this order. SEs and typical enhancers from the same cell type as the TF ChIP-seq were used. Promoters were defined as 4 kb windows centered on the transcription start sites of RefSeq genes. Gene bodies were defined as the remaining portion of genes longer than 2 kb from 2 kb downstream of the promoter to the transcription end site.

**SE identification and assignment.** SEs were identified using ROSE (https://bitbucket.org/young_computation/rose) algorithm with some modifications. Reads overlapping ENCODE blacklist regions were removed using bedtools intersect. From the remaining reads, two sets of peaks of H3K27ac were identified using MACS with parameter sets –keep-dup=auto –p 1e−9 and –keep-dup=all –p 1e−9. H3K27ac peaks were stitched computationally if they were within 12500 bp of each other, though peaks fully contained within+/− 2000bp from a RefSeq promoter were excluded from stitching. These stitched enhancers were ranked by their H3K27ac signal (length * density) with input signal subtracted. SEs were defined geometrically as those enhancers above the point at which the line $y = x$ is tangent to the curve. Stitched enhancers (typical enhancers and SEs) were assigned to the single active gene whose transcription start site is nearest the center of the stitched enhancer. Active genes were determined by taking the top two-thirds of all RefSeq promoters (+/−500 bp) ranked by their H3K27ac signal. H3K27ac signal in promoters was determined using bamToGFF (https://github.com/BradnerLab/pipeline) with parameters –e 200 –m 1 –r –d.

**RNA-seq and bioinformatic analysis.** RNA isolation was performed using the RNeasy Mini Kit (Qiagen) and RNA quality was assessed on a Fragment Analyzer (Advanced Analytical Technologies) − SS Total RNA 15nt. RNA-seq libraries were prepared using the NEBNext Ultra II Directional RNA Library Prep Kit for Illumina (New England Biolabs) with Poly(A) selection according to the manufacturer's instructions. Library quantification was examined on a Fragment Analyzer − HS NGS Fragment 1-6000 bp and Qubit HS dsDNA Kit (Invitrogen). Libraries were pooled and sequenced to 75 bp single-end on the Illumina NextSeq 500 platform. Sequencing data were analyzed as described[60].

**CRISPR/Cas9 genome editing.** Cloning of gRNA lentiCRISPR v2 plasmid, lentiviral packaging, lentiviral transduction and single clone isolation were performed as described[12]. The lentiviral Cas9-containing plasmid lentiCRISPR v2[61] was a gift from F. Zhang (Addgene #52961, Cambridge, USA). Non-targeting and *BATF3* gRNAs were designed using the E-CRISP program, version 5.4 (www.e-crisp.org/E-CRISP/index.html), which targeting the first exon of *BATF3*. Sequences of gRNA oligonucleotides are listed in Supplementary Table 8.

**Immunostaining and quantification.** IHC was performed on FFPE (formalin-fixed paraffin embedded) sections with the conventional avidin–biotin–peroxidase method. Heat antigen retrieval was performed using citrate buffer pH 6.1 (Agilent Dako). Endogenous peroxidases were quenched by incubating sections in 3% $H_2O_2$ in PBS. Sections were blocked using Avidin/Biotin Blocking Kit (Vector Laboratories). Primary antibodies were added in 1% BSA/PBS at 4 °C overnight. Slides were incubated with biotin-conjugated secondary antibodies using IDtect Super Stain System—HRP (Empire Genomics) and developed using AEC Substrate Kit (BDPharmingen). Slides were mounted with Aquatex (Merck). Primary antibodies used are listed in Supplementary Table 7 and dilutions used are as follows: CD30 (1:50), IL-2 (1:200), IL-2Rα (1:50), IL-2Rβ (1:40), IL-2Rγ (1:20), BATF3 (100 µg/ml), IL-15 (10 µg/ml) and IL-15Rα (7 µg/ml). Stained slides were further assessed by two experienced pathologists who were blinded to the clinicopathological parameters and patient outcome. Quantification of the slides was determined using the histoscore system. Stained slides were scored for both the intensity and percentage of positively staining cells: 0 negative, 1+ weak, 2+ moderate and 3+ strong staining. Positive staining was considered if present in >1% of cells. The Histoscore was then calculated by multiplying the intensity by the percentage of positively stained cells.

**End-point and quantitative RT-PCR.** RNA preparation, cDNA synthesis, semi-quantitative and quantitative PCR analyses were performed as described[9]. Primers are listed in Supplementary Table 8.

**Flow cytometry.** Surface protein expression of IL-2Rα, IL-2Rβ, IL-2Rγ, CD30 and IL-15Rα on ALCL and T-cell leukemia-derived cell lines was determined by flow cytometry. Primary (all 1:50) and corresponding isotype control (all 1:50) antibodies are listed in Supplementary Table 7. Apoptotic cell death was determined by Annexin V/PI staining according to the manufacturer's instructions (Bender MedSystems). Data were acquired with a BD FACSCanto II and analyzed with FACSDiva software 8.0.1 (Becton Dickinson).

**EMSA.** Nuclear protein preparation and EMSA were performed as described[9]. AP-1 motifs of the respective *IL2R* promoter (*IL2RA*) or enhancer (*IL2RB*) gene regions were identified using LASAGNA-Search 2.0 online tool and EMSA probes

were designed accordingly. EMSA oligonucleotides are listed in Supplementary Table 8. Based on the experimental method, no molecular weight markers can be shown for EMSA analyses.

**ChIP and real-time PCR analyses**. ChIP assays were performed in two biological replicates using antibodies for BATF3 (AF7437, R&D Systems) according to a modified Millipore protocol (http://www.merckmillipore.com/DE/de/product/ChromatinImmunoprecipitation%28ChIP%29-Array-Kit,MM_NF-17-295#anchor_BRO) as described[12]. In short, cells were fixed using the two-step cross-linking method (2 mM disuccinimidyl glutarate for 30 min followed by 1% formaldehyde for 5 min) and lyzed with 50 mM Tris-HCl, pH 8.0/5 mM EDTA/1% SDS. Thereafter, samples were sonicated with the Bioruptor Plus (Diagenode; 13 cycles, intensity High, sonication 30 s/break 30 s per cycle). Chromatin was pre-cleared with bovine serum albumin (BSA)-saturated Protein A-Sepharose and incubated overnight at 4 °C with BATF3 antibody (27 µg/1 × 10$^7$ cells). Immuno-complexes were collected with BSA-saturated Protein A-Sepharose for 1 h at 4 °C. After washing, protein-DNA complexes were eluted using 1% SDS/0.1 M NaHCO$_3$. Reversal of the cross-linking, RNAse treatment, proteinase K digestion, and DNA purification by phenol-chloroform extraction were performed according to standard protocols. Primary antibody and primer sequences are shown in Supplementary Table 7 and Table 8, respectively. Quantitative PCR was carried out in technical triplicates with ChIP-DNA isolated from to 3 × 10$^5$ cell equivalents using the CFX96 system and GoTaq qPCR Master Mix (Promega). Non-recruiting intergenic regions on chromosomes 4 (Ref_Ch4) were amplified as references. A total of 4 ng input DNA was used as a control.

**Cell proliferation assay**. Cells were starved in medium with 1% FCS overnight and seeded in 96-well plates with 2 × 10$^4$ cells per well. Cell proliferation was measured after 72 h of cytokine stimulation with or without monoclonal antibody treatment by resazurin assay or [$^3$H]-thymidine incorporation according to manufacturer's instructions. All values were normalized to the control-treated cells.

**Immunoblotting**. Whole protein lysate preparation and immunoblot analyses were performed as described[28]. Primary antibodies used are listed in Supplementary Table 7 and dilutions used are as follows: BATF3 (1 µg/ml), STAT1 (1:1000), pSTAT1 (Y701) (1:1000), STAT3 (1:1000), pSTAT3 (Y705) (1:1000), STAT5 (1:1000), pSTAT5 (Y694) (1:1000), ERK1/2 (1:1000), pERK1/2 (T202/Y204) (1:1000), PARP1 (1:1000), β-ACTIN (1:1000), β-TUBULIN (1:1000) and GAPDH (1:10000).

**Murine xenograft model**. Both male and female NSG mice were purchased from the National Cancer Institute (Frederick, MD). ALCL cells (1 × 10$^6$; FE-PD, Mac-2A or TLBR-1) were subcutaneously implanted into the hind flanks of nine- to ten-week old mice. When tumor volumes reached ~100–150 mm$^3$, mice were randomly allocated into three groups to receive vehicle (PBS), non-binding ADC (B12-SG3249) or ADCT-301 (camidanlumab tesirine) (both ADC Therapeutics, 0.5 mg/kg, intravenous injection). Tumors were measured with caliper every three days. Tumor size was estimated using the following formula: volume (mm$^3$) = (length × width$^2$)/2. Each mouse was euthanized when its tumor reached the end-point volume of 2000 mm$^3$ or at the study end. Animal work was carried out according to an ethical animal license protocol that was approved by the Medical University of Vienna and Austrian Ministry of Education and Science (BMBWF-66.009/0200-V/3b/2018 and Addendum Zl. 18/115-97/98, 2019).

**Statistical analysis**. Statistical analyses were performed using GraphPad Prism v6.0 and R v4.0.3 software. Data of protein expression in TMAs are presented in box-and-whisker plots with indication of medians, quartiles and ranges, each data point represents an individual score. Correlation of data was determined by the Pearson correlation coefficient. Other data are presented as means ± SD or SEM as stated in the respective figure legends. Differences between mean values of two groups were evaluated by the two-tailed unpaired Student's $t$ test. Estimation of EFS and OS was performed using the Kaplan–Meier method using IBM SPSS Statistics Version 22 with the survival package of R (v3.2-7; Therneau, 2020). Events were defined as relapse, progressive disease or death from any cause. Differences between groups were compared by a log-rank test. $P$ values < 0.05 were considered to be statistically significant.

**Reporting summary**. Further information on research design is available in the Nature Research Reporting Summary linked to this article.

## Data availability

All RNA-seq and ChIP-seq datasets produced in this study are deposited in Gene Expression Omnibus (GEO) under SuperSeries GSE158916. Source data is provided for the Fig. 1a–d, Supplementary Fig. 1 and Supplementary Fig. 2a–d. The Crescenzo et al.[29] and Iqbal et al.[30] publicly available data used in this study are available in the Sequence Read Archive (SRA) and GEO database under accession code SRP044708 and GSE19069, respectively. ChIP-seq of H3K27ac and input DNA of normal T-cell subsets and Jurkat

cell line used in this study are publicly available in GEO under the accession numbers GSM1058764, GSM1058789, GSM772835, GSM772916, GSM1102781, GSM1102806, GSM1296384 and GSM569086. The remaining data are available within the Article, Supplementary Information or Source Data file. Source data are provided with this paper.

## Code availability

Computer code used in this study is available within Supplementary Software.

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

## Acknowledgements

We thank Boris Tichý (Brno) at the Core Facility Genomics of CEITEC Masaryk University (LM2018132), Eva Kaergel and Simone Lusatis (both Berlin) for their outstanding technical assistance. We also thank Gerda Egger, Heidi Neubauer, Inés Garces de los Fayos Alonso, Philipp Staber, Renato Arnese, Richard Moriggl, Robert Eferl, Sabine Lagger (all Vienna) for helpful discussions. We are grateful to Alan Epstein (Los Angeles) for providing the BIA-ALCL cell lines. We also thank ADC Therapeutics for providing ADCT-301 and B12-SG3249. This project received funding from the European Union Horizon 2020 Marie Skłodowska-Curie Innovative Training Networks (ITN-ETN) under grant agreement no. 675712. O.M. is supported by the Austrian Science Fund (FWF) project (P32579). B.J.A. receives support from the American Lebanese Syrian Associated Charities. D.S. received funding from FWF grant P32693. Š.P. is financially supported by the Ministry of Education, Youth and Sports of the Czech Republic under the project CEITEC 2020 (LQ1601). W.K. is supported by the Kinder Krebs Initiative Buchholz-Holm-Seppensen (KKI). L.K. was supported by the Christian-Doppler Lab for Applied Metabolomics (CDL-AM). L.K. was supported by the Austrian Science Fund, grants FWF grant P 26011 and P 29251 and by the COMET Competence Center CBmed - Center for Biomarker Research in Medicine (FA791A0906.FFG). The COMET Competence Center CBmed is funded by the Austrian Federal Ministry for Transport, Innovation and Technology (BMVIT); the Austrian Federal Ministry for Digital and Economic Affairs (BMDW); Land Steiermark (Department 12, Business and Innovation); the Styrian Business Promotion Agency (SFG); and the Vienna Business Agency. The COMET program is executed by the FFG. L.K. was recipient of funds from a European Union Horizon 2020 Marie Sklodowska-Curie Innovative Training Network (ITNETN) grant, award n. 675712.

## Author contributions

H.-C.L., S.M. and O.M. conceived and designed the research. H.-C.L., M.C., E.G., S.S., I.A. and N.S. acquired data. N.P., M.W.Z., B.J.A. and A.T.L. performed genome-wide H3K27ac occupancy analysis. I.A.M.-M., S.T., F.F. and L.K. reviewed and quantified IHC staining. C.L., J.O., T.H. and Š.P. performed RNA-seq and bioinformatics analysis. A.J., N.P., S.Pe., W.K., F.K., C.D.-W., W.W., I.S.-K., L.C., S.Pi., L.L., D.Si., V.F., P.G. and C.A. provided TMAs and clinical data collection and analysis. F.Z. provided ADCT-301 and B12-SG3249. D.-A.S., K.C.G. and D.B. synthesized and provided Neo-2/15. J.W. and D.St. performed xenograft experiments. S.D.T. provided critical reagents. H.-C.L. and O.M. drafted the manuscript. M.J., S.D.T., S.M. and L.K. made critical revisions of the manuscript. All authors discussed the results and commented on the manuscript.

## Competing interests

B.J.A. is a shareholder in Syros Pharmaceuticals. F.Z. is a shareholder and employed by ADC Therapeutics. The other authors declare that they have no conflict of interest.

## Additional information

[1]Department of Pathology, Unit of Experimental and Laboratory Animal Pathology, Medical University of Vienna, Vienna, Austria. [2]European Research Initiative on ALK-Related Malignancies (ERIA), Suzanne Turner, Cambridge, UK. [3]Group Biology of Malignant Lymphomas, Max-Delbrück-Center (MDC) for Molecular Medicine, Berlin, Germany. [4]Department of Hematology, Oncology, and Cancer Immunology, Charité—Universitätsmedizin Berlin, corporate member of Freie Universität Berlin and Humboldt-Universität zu Berlin, Berlin, Germany, and Experimental and Clinical Research Center (ECRC), a joint cooperation between the MDC and Charité, Berlin, Germany. [5]Department of Pediatric Oncology, Dana-Farber Cancer Institute, Harvard Medical School, Boston, MA, USA. [6]Institute of Pathology and Neuropathology, University Hospital and Comprehensive Cancer Center Tübingen, Tübingen, Germany. [7]Department of Computational Biology, St. Jude Children's Research Hospital, Memphis, TN, USA. [8]Division of Cellular and Molecular Pathology, Department of Pathology, University of Cambridge, Addenbrooke's Hospital, Cambridge, UK. [9]Christian Doppler Laboratory (CDL) for Applied Metabolomics, Medical University of Vienna, Vienna, Austria. [10]Unit of Laboratory Animal Pathology, University of Veterinary Medicine Vienna, Vienna, Austria. [11]Central European Institute of Technology (CEITEC), Masaryk University, Brno, Czech Republic. [12]Institute of Pathology, University of Würzburg, Würzburg, Germany. [13]German Cancer Consortium (DKTK) German Cancer Research Center (DKFZ), Heidelberg, Germany. [14]Department of Pathology and Laboratory Medicine, Aga Khan University Hospital, Karachi, Pakistan. [15]Department of Pathology, Hematopathology Section, University Hospital Schleswig-Holstein Campus Kiel, Kiel, Germany. [16]ADC Therapeutics (UK) Limited, London, UK. [17]Institute for Protein Design, University of Washington, Seattle, WA, USA. [18]Department of Biochemistry, University of Washington, Seattle, WA, USA. [19]Division of Life Science, The Hong Kong University of Science and Technology, Kowloon, Hong Kong. [20]Departments of Molecular and Cellular Physiology and Structural Biology, Stanford University School of Medicine, Stanford, CA, USA. [21]Howard Hughes Medical Institute, Chevy Chase, MD, USA. [22]Department of Internal Medicine—Hematology and Oncology, University Hospital Brno, Brno, Czech Republic. [23]Department of Pharmacology, Physiology and Microbiology, Division Pharmacology, Karl Landsteiner University of Health Sciences, Krems, Austria. [24]Department of Dermatology, Medical University of Graz, Graz, Austria. [25]Division of Haematopathology, European Institute of Oncology IRCCS, Milan, Italy. [26]Institute of Pathology, Lausanne University Hospital (CHUV) and Lausanne University, Lausanne, Switzerland. [27]Hematology Department, Necker University Hospital, Assistance Publique—Hôpitaux de Paris, and Institut Necker-Enfants Malades, INSERM UMR1151 (Normal and pathological lymphoid differentiation), Université de Paris, Paris, France. [28]Department of Pathology, Henri Mondor University Hospital, AP-HP, INSERM U955, University Paris East, Créteil, France. [29]Department of Dermatology, HELIOS Hospital Krefeld, Krefeld, Department of Dermatology and Allergy, Charité—Universitätsmedizin Berlin, Berlin, Germany. [30]Pediatric Hematology and Oncology, University Hospital Hamburg-Eppendorf, Hamburg, Germany. [31]Center for Biomarker Research in Medicine (CBMed) Core Lab 2, Medical University of Vienna, Vienna, Austria. [32]These authors contributed equally: Stephan Mathas, Lukas Kenner, Olaf Merkel. ✉email: stephan.mathas@charite.de; lukas.kenner@meduniwien.ac.at; olaf.merkel@meduniwien.ac.at

