## [Peer Review File · Nature Communications]

Super-enhancer-based identification of a BATF3/IL-2R-module reveals vulnerabilities in anaplastic large cell lymphomaReviewers' Comments:

Reviewer #1:

Remarks to the Author:

In this manuscript, Liang et al. identify superenhancers in Anaplastic large cell lymphoma (ALCL), an aggressive CD30-positive T-cell lymphoma. They elucidate a link between BATF3, IL-2 receptor and STAT1, STAT5 and ERK1/2. They also show that IL-2R α -expression in ALCL patients is associated with more aggressive cancers, and most excitingly, demonstrate that an IL-2R α -targeting antibody-drug conjugate will kill ALCL cells in cell lines and in vivo. The manuscript is largely well written with well prepared figures, but I have some major issues about the conclusions that the authors are trying to draw:

1. First, it would be helpful if Liang et al could draw out a proposed schematic of their mechanism, particularly with regard to the link between BATF3, IL-2 receptor, STAT1, STAT5, and ERK1/2 because the mechanism is a bit vague. My guess is that Liang et al. want to say that BATF3 binds to a superenhancer at IL2RA gene (maybe IL2RB and IL2RG too), which leads (somehow) to STAT1, STAT5, and ERK1/2 upregulation, and this in turn promotes ALCL growth, and targeting IL2R α protein via the antibody will inhibit this pathway, and thus lead to reduced ALCL growth. Liang et al. also mention CD30 and JUNB as players in this process, but I am unclear how they fit into this mechanism.

2. A major limitation of this work is that the superenhancer profiling was only done with cell lines and only two cell lines - Karpas-299 and Mac-1. It would have been helpful if some ALCL clinical samples and normal cells could have been performed. Multiple clinical samples would allow for understanding as to how frequently, and in which samples, the superenhancer at IL2RA and IL2RB can be seen. This would then allow for investigation as to whether normal T cells have superenhancers at BATF3 and IL2ra, which would be useful to know in understanding potential side effects of the ADCT-301 on normal T cells.

3. At the start of the manuscript, in line 71-72, the authors suggest that a thymic origin has been proposed for ALCL, making it seem as if the manuscript will try to resolve this question, however, this question does not seem to be really tackled or answered by Liang et al. Perhaps this could be addressed or if not, this introduction should perhaps be rewritten so as to focus on the central questions explored by this paper.

3. Jurkat is IL2R α negative. Does show an absence of a superenhancer at the IL2R α gene promoter?

4. CRISPR is used to delete BATF3 followed by RNA-Seq in Figure 1d. However, CRISPR can have off-target effects, and if only one clone was studied, the downstream effects identified could potentially be due to off-target effects. Liang et al should test 3 different clones by RT-qPCR to see if they all show IL2RG downregulation.

5. I am confused about the relationship between IL2RA, IL2RB, and IL2RG. In figure 1a, Liang et al show that IL2RA and B are marked by superenhancers, but IL2RAG is not marked by superenhancers. However, in Supp. Figure 1b, Liang et al show that BATF deletion leads to significant IL2RG downregulation but not significant IL2RA and IL2RB. So I am not sure what the authors are trying to say here about the connection between BATF3 and the different IL-2R subunits. Also, 1e shows BATF3 correlated genes but I only see IL2RA and IL2RB and not IL2RG. In figure 1f, IL2R α is examined but not IL2RB and IL2RG.

6. Is there a BATF3 auto-regulatory loop - does BATF3 bind to the superenhancer at BATF3? Superenhancers are usually thought to auto-regulate.

7. Do the BATF3 CRISPR deleted cells show altered growth? Do the BATF3 CRISPR deleted cells show changes to STAT1, STAT5 and ERK1/2? And can such changes be rescued back by BATF3

overexpression?

8. A number of cell lines were tested for IL2R expression in Fig 2a. How about BATF3 levels in these cell lines? Why was BATF3 only tested in a few of these cell lines but not all?

9. Is there an AP-1/TRE site at the CD30 regulatory regions?

10. Fig 3 - does IL2RA knockdown reduce ALCL proliferation? Does it lead to rescue of the activation of STAT1, STAT5 and ERK1/2?

11. How do the BATF3 CRISPR deleted cells respond to rhIL-2 stimulation as done in Fig. 3b? Assuming ILR2A/B/G are downregulated by BATF3 CRISPR deletion, my guess would be that these cells should show no response to rhIL-2 and there would be no growth promoting effect.

12. The pediatric ALCL experiment with IL2RA is interesting. How about IL2RB and IL2RG? Why were these not studied?

13. In the ALCL cell lines treated with ADCT-301, what happens to STAT1 and STAT5 levels? Do they get reduced? Are these levels unchanged in the unaffected Jurkat and KE-37 cells? In the ALCL cell lines, are there differences in ILR2A levels and do these different levels lead to different rates of cell killing with ADCT-301?

14. Would ADCT-301 also affect normal T cells, since these potentially also express ILR2A?

Reviewer #2:

Remarks to the Author:

1) Major claims of the paper

This is an interesting paper showing that BATF3 and IL2R chains are probably key regulators for the oncogenic activity in ALCL. It also shows that targeted inhibition of IL2Ra with an antibody-drug conjugate, effectively kills ALCL cells in vitro and in vivo making it a promising strategy in (relapsed/refractory) ALCL.

2) Are the findings novel and will they be of interest in the community and in the field?

The biological findings described in this study are novel and will, if confirmed, have the potential to advance the field in terms of both biological insight and new therapeutic options for patients with ALCL. However, in contrast with some of the introductory and final statements in the manuscript, ALCL histology, in particular ALK+ ALCL, primary cutaneous (ALK-)ALCL and BIA-ALCL, is recognized as being associated with a significantly better prognosis than other aggressive T-cell lymphomas. Recently, biological information has improved the prognostic granularity within the largest subset of ALCL, i.e. adult systemic ALCL. Therefore it would be useful to have this information for the presented cohort in order to fully understand the clinical potential of the reported findings in ALCL. The ongoing NCT02432235 trial is currently evaluating the targeting of IL2Ra in different types of lymphoma, including those of T-cell origin.

3) Reproducibility of the presented work? The methodology of the study is thoroughly explained and should allow other groups to try to reproduce results in independent cohorts

4) Is further evidence required to strengthen the conclusions?

The authors state in different occasions that there is an unmet clinical need for patients with ALCL type lymphoma. While it may be applicable in a subset of systemic ALK-neg ALCL, this statement is (as mentioned under pt.2) not entirely correct for all cases of ALK-neg ALCL. Recent reports (Parilla-Catellar et al, Blood 2014; Pedersen MB et al, Blood 2017) have shown that additional biological

features, such as the presence of rearrangements in the DUSP22 or the TP63 gene are associated (under current treatment approaches) with a favourable or unfavourable prognosis, respectively. To show unequivocal efficacy of IL2Ra inhibition in the setting of TP63 rearranged ALK-neg systemic ALCL, would undoubtedly strengthen the conclusions of this paper.

5) Any further question deemed relevant in this context?

(i) In the introductory sentence of the Abstract, the authors give an overview of ALCL subtypes. The primary cutaneous (consistently ALK-neg) ALCL is missing. It should be mentioned as an entity with good prognosis despite of ALK-negativity.

() page 4, line 24: It would further strengthen the biologic observation of this study, if the authors could show that also cases of primary cutaneous ALCL display a strong expression of IL2R chains and/or BATF3 similarly to the other ALCL subgroups. In fact, suppl.Table 3 shows that the primary tissue biopsy used for TMA construction in this study was 'SKIN' in 6 (10%) of the tabulated 60 ALK-neg ALCL pts. Could it be possible that some of these cases were primary cutaneous ALCLs?

(i) page 6, line 14: The pts from the NHL-BFM and the ALCL99 studies were treated in trials run 20 and 10 yrs ago, respectively, i.e. prior to the advent of brentuximab vedotin. The outcomes for both ALK+ ALCL subsets (IL2Ra low and IL2R high) shown in Fig.4a are high (best group 100% OS at 5 yrs; worst group 80% OS at 5 yrs). Not exactly an 'unmet clinical need', at least in the first-line setting. Moreover, one could assume that the addition of brentuximab vedotin to the treatment schedule could improve outcome also for pts with high IL2Ra levels to values above 80% 5-yr OS. This should be mentioned/discussed.

(ii) The patient cohort of adult ALK-neg ALCL pts shown in Fig.4a is very small, i.e. 15 pts subdivided in 2 groups: 5 pts with low IL2Ra expression (better outcome) and 10 with high IL2Ra expression (worse outcome). A larger cohort size would allow for more robust conclusions.

What was the DUSP22 and TP63 rearrangement status of these pts? What was the individual treatment background of these patients? Were there any cases with primary skin involvement in the IL2Ra 'high' group?

(iii) page 7, line 32 one line from the bottom): Alk-inhibitors (e.g. crizotinib) should be mentioned in the context of ALK+ ALCL (e.g. Mossè YP, JCO 2017).

(iv) page 8, line 1: The ECHELON-2 study mentioned did not (was not powered to) demonstrate superiority in other entities than ALCL. The effect of brentuximab vedotin in the trial was most pronounced in ALK+ than ALK-neg ALCL. Also, a subset analysis performed on the ECHELON-2 dataset (Savage K et al, ASH 2019, oral presentation) showed, in ALK-ALCL pts, a possible benefit of autologous transplant consolidation in BV-treated ALCL pts. No information on DUSP22 and TP63 status was available from the ECHELON-2 data set.

(v) page 8, line 9 (last sentence): This point would be significantly strengthened, if the authors could demonstrate efficacy in TP63-rearranged ALK-neg ALCL

Reviewer #3:

Remarks to the Author:

The authors use analysis of superenhancers in cell lines to identify a regulatory module of BATF3 with IL2 receptor genes that facilitates IL2/IL15-mediated activation of STATs, enhances cell growth, and can be targeted with an anti-IL2RA ADC. The studies are well conducted and the overall findings convincing.

1. The authors should temper their statement that the "prognosis [of ALCL] is unsatisfactory" with data on the often good prognosis in ALK+ ALCL, some subsets of ALK- ALCL (especially in BV era), and cutaneous ALCL, which is not mentioned at all and should be addressed. Also, they need to acknowledge that Mac-1, 1 of 2 lines on which the original SE discovery was performed, is not a true ALCL cell line, having been derived from Sezary-like cells from the peripheral blood of a patient with lymphomatoid papulosis and cutaneous T-cell lymphoma who only later progressed to large cell lymphoma (PMID: 1532439). It is also somewhat overstated that the BATF3/IL2R module is a "new

vulnerability" in ALCL since only IL2R (and not BATF3) is proposed as a direct target, and the expression of IL2R in ALCL and its potential as a target have been known for some time.

2. The hypothesis "that IL2 might originate from the tumor environment" is not adequately tested by the methodology and the image in Fig. 3a is insufficient to justify this. This should either be removed or replaced by quantitative and properly controlled experiments to test this hypothesis.

Below you find our detailed answers to the reviewer questions:

Reviewer #1, expert in super-enhancers and epigenomics (Remarks to the Author):

In this manuscript, Liang et al. identify superenhancers in Anaplastic large cell lymphoma (ALCL), an aggressive CD30-positive T-cell lymphoma. They elucidate a link between BATF3, IL-2 receptor and STAT1, STAT5 and ERK1/2. They also show that IL-2R α -expression in ALCL patients is associated with more aggressive cancers, and most excitingly, demonstrate that an IL-2R α -targeting antibody-drug conjugate will kill ALCL cells in cell lines and in vivo. The manuscript is largely well written with well prepared figures, but I have some major issues about the conclusions that the authors are trying to draw:

1. First, it would be helpful if Liang et al could draw out a proposed schematic of their mechanism, particularly with regard to the link between BATF3, IL-2 receptor, STAT1, STAT5, and ERK1/2 because the mechanism is a bit vague. My guess is that Liang et al. want to say that BATF3 binds to a superenhancer at IL2RA gene (maybe IL2RB and IL2G too), which leads (somehow) to STAT1, STAT5, and ERK1/2 upregulation, and this in turn promotes ALCL growth, and targeting IL2Ra protein via the antibody will inhibit this pathway, and thus lead to reduced ALCL growth. Liang et al. also mention CD30 and JUNB as players in this process, but I am unclear how they fit into this mechanism.

We thank the reviewer for his suggestion and have now included a scheme which visualizes the proposed mechanism (**NEW Fig. 4e**).

2. A major limitation of this work is that the superenhancer profiling was only done with cell lines and only two cell lines - Karpas-299 and Mac-1. It would have been helpful if some ALCL clinical samples and normal cells could have been performed. Multiple clinical samples would allow for understanding as to how frequently, and in which samples, the superenhancer at IL2RA and IL2B can be seen. This would then allow for investigation as to whether normal T cells have superenhancers at BATF3 and IL2ra, which would be useful to know in understanding potential side effects of the ADCT-301 on normal T cells.

To address this question, we have now performed SE profiling of 6 additional ALCL cell lines (4 ALCL, ALK⁺ cell lines: SU-DHL-1, SR-786, JB6, SUP-M2; 2 ALCL, ALK⁻ cell lines: Mac-2A and FE-PD), 2 primary clinical samples of ALCL-affected lymph nodes from ALCL patients and, in response to the question on normal T cells, analyzed publicly available data sets of CD3, CD4 and CD8 T cells from healthy donors. Analyses confirmed the presence of SEs at *BATF3*, *IL2RA* and *IL2RB* loci as a common feature in ALCL cells, whereas in the normal T-cell subsets no SE formation was observed. These results are in full support of our previous conclusions. These data are now included in **NEW Fig. 1d**, **NEW Supplementary Fig. 1a, b, c**.

It is very challenging to find fresh frozen, viable ALCL patient samples in sufficient cell numbers to perform ChIP-seq analyses (20 million cells per ChIP-seq are at least required). Despite this, we were able to successfully analyse 2 primary patient samples using the cut-and-run ChIP-seq technique that works with lower cell numbers. In both samples, ChIP-seq tracks could be retrieved and SE analysis could be performed. Hockey-stick plots of these two patients are now to be found in **NEW Fig. 1d** and ChIP-seq tracks in **NEW Supplementary Fig. 1c**, again confirming SE at *BATF3*, *IL2RA* and *IL2RB* loci as a common ALCL feature.

3. At the start of the manuscript, in line 71-72, the authors suggest that a thymic origin has been proposed for ALCL, making it seem as if the manuscript will try to resolve this question, however, this question does not seem to be really tackled or answered by Liang et al. Perhaps

this could be addressed or if not, this introduction should perhaps be rewritten so as to focus on the central questions explored by this paper.

The introduction has been modified following the reviewer's suggestion.

3. Jurkat is IL2Ra negative. Does show an absence of a super enhancer at the IL2Ra gene promoter?

We have retrieved and analyzed Jurkat H3K27ac ChIP-seq data from public databases. There is no SE of the *IL2RA* gene in Jurkat cells. These data are now included in the manuscript (see **NEW Supplementary Fig. 1b**).

4. CRISPR is used to delete BATF3 followed by RNA-Seq in Figure 1d. However, CRISPR can have off-target effects, and if only one clone was studied, the downstream effects identified could potentially be due to off-target effects. Liang et al should test 3 different clones by RT-qPCR to see if they all show IL2RG downregulation.

We thank the reviewer for his/her excellent suggestion to avoid the analysis of clone-specific off-target effects. However, we already had included 3 different clones from the Karpas-299 cell line for our initial RNA-seq analyses. The BATF3 immunoblotting of the different clones is now included in the **NEW Supplementary Fig. 2a** and specified in the text of the results section. All 3 *BATF3*-KO clones showed downregulation of *IL2RG*.

5. I am confused about the relationship between IL2RA, IL2RB, and IL2RG. In figure 1a, Liang et al show that IL2RA and B are marked by superenhancers, but IL2RAG is not marked by superenhancers. However, in Supp. Figure 1b, Liang et al show that BATF deletion leads to significant IL2RG downregulation but not significant IL2RA and IL2RB. So I am not sure what the authors are trying to say here about the connection between BATF3 and the different IL-2R subunits. Also, 1e shows BATF3 correlated genes but I only see IL2RA and IL2RB and not IL2RG. In figure 1f, IL2Ra is examined but not IL2RB and IL2RG.

To address the reviewer's legitimate concerns, we have reworded relevant phrases in the results section to make the manuscript clearer on these points. Overall, we can state that our data now provide robust evidence that *IL2R* components, in particular *IL2RA* and *IL2RB*, are consistently found among the top hits in the SE analyses across all the cell lines (**NEW Supplementary Fig. 1 a, b**) and also in primary ALCL cells (**NEW Fig. 1d**), that IL-2R components are commonly and strongly deregulated in ALCL (**Fig. 1h and Fig. 2a**), that in particular IL-2R α correlates with BATF3 expression (**Fig. 1f, g and Supplementary Fig. 2e, f**), and that all tested 9 ALCL cell lines can be efficiently killed by IL-2R α -targeting (**Fig. 4c**). We have to state here that *IL2RG* was very strongly (>30 fold) downregulated following *BATF3* deletion in Karpas-299; however, also *IL2RA* and *IL2RB* were downregulated, but only 2-3 fold (see **Fig. 2b**).

6. Is there a BATF3 auto-regulatory loop - does BATF3 bind to the super-enhancer at BATF3? Super-enhancers are usually thought to auto-regulate.

To answer this interesting question, we performed genome-wide BATF3-ChIP analyses in Karpas-299 and Mac-1 cells and find binding of BATF3 to its own locus in both cell lines, which indeed implies autoregulation. These data are included in **NEW Supplementary Fig. 2d** and this finding is now stated in the results section of the manuscript.

7. Do the BATF3 CRISPR deleted cells show altered growth? Do the BATF3 CRISPR deleted cells show changes to STAT1, STAT5 and ERK1/2? And can such changes be rescued back by BATF3 overexpression?

We have previously demonstrated that loss or downregulation of BATF3 results in reduced proliferation and viability in Karpas-299 cells and Mac-1 cells [1]. To answer the question of the reviewer in more depth, we also generated CRISPR/Cas9-mediated *BATF3*-KO cells of Karpas-299, SUP-M2 and Mac-2A cell lines. In all these cell lines, *BATF3* CRISPR-deletion results in growth retardation, which can be rescued by BATF3 ectopic expression. We did not observe consistent alterations of STAT1, STAT5 or ERK1/2 expression or activity following *BATF3* deletion (**NEW Supplementary Fig. 3h, i**).

8. A number of cell lines were tested for IL2R expression in Fig 2a. How about BATF3 levels in these cell lines? Why was BATF3 only tested in a few of these cell lines but not all?

To address this question, we have now monitored both mRNA and protein expression of BATF3 in the cell lines used in **Fig. 2a**. We see strong expression of BATF3 in all ALCL cell lines but not in control cell lines. These data are now included in the manuscript (**NEW Supplementary Fig. 3d**). Immunoblotting for several of the cell lines can also be found in Schleussner *et al.*, *Leukemia* 2018, 32:1994 [1].

0. Is there an AP-1/TRE site at the CD30 regulatory regions?

Yes, AP-1/TRE sites have been described in the promoter region of *CD30* (Watanabe, M. *et al. Am J Pathol*, 2003[2]; Watanabe, M. *et al. Cancer Res*, 2005[3]). The respective citations are now included in the manuscript.

1. Fig 3 - does IL2RA knockdown reduce ALCL proliferation? Does it lead to rescue of the activation of STAT1, STAT5 and ERK1/2?

Our data suggest that IL-2R α is not constitutively active *per se*, but instead requires the exogenous addition of IL-2 mediating the growth promoting effect on ALCL cells. We have shown that the addition of IL-2 stimulates the growth of ALCL cells, and this effect can be specifically blocked by the IL-2R α -inhibiting antibody basiliximab (**Fig. 3e**). Similarly, addition of basiliximab blocks at least in part IL-2-induced pSTAT5 and pERK1/2-signaling (**Supplementary Fig. 4g**). Thus, we did not perform an *IL2RA*-knockdown, as we did not expect an effect on ALCL cell growth in the absence of IL-2. In line, data from the DepMap consortium show that *IL2RA* knockdown is not killing ALCL cell lines. (<https://depmap.org/portal/interactive/>).

2. How do the BATF3 CRISPR deleted cells respond to rhIL-2 stimulation as done in Fig. 3b? Assuming IL2A/B/G are downregulated by BATF3 CRISPR deletion, my guess would be that these cells should show no response to rhIL-2 and there would be no growth promoting effect.

We have performed the proposed experiment and indeed we do see no growth stimulation by IL-2 or its mimic Neo-2/15 in *BATF3*-deleted cells. These data are now included in **NEW Fig. 3b and NEW Supplementary Fig. 4d**.

3. The pediatric ALCL experiment with IL2RA is interesting. How about IL2RB and IL2RG? Why were these not studied?

As now further supported by our extended SE analyses, we focused in our work mainly on IL-2R α and IL-2R13, for which we provide the strongest evidence on consistent expression in ALCL and a regulatory link to BATF3 expression. Therefore, we did also analyze IL-2R13 (but not IL-2R γ) in this context, but there was no significant association to overall survival. If requested by the reviewer we could also include these data in the manuscript.

4. In the ALCL cell lines treated with ADCT-301, what happens to STAT1 and STAT5 levels? Do they get reduced? Are these levels unchanged in the unaffected Jurkat and KE-37 cells?

In the ALCL cell lines, are there differences in ILR2A levels and do these different levels lead to different rates of cell killing with ADCT-301?

As suggested by the reviewer, we treated cell lines Karapas-299, SR-786 (both ALK⁺) and FE-PD (ALK⁻) with ADCT-301 or the corresponding control antibody B12-SG3249, and then analyzed STAT1 and STAT5 levels and phosphorylation. In all the ALCL cell lines, we could detect moderate decreases of pYSTAT5 levels, and in FE-PD cells also pYSTAT1 was decreased. No such effects were observed following treatment with the control ADC B12-SG3249. In Jurkat cells, we could not detect relevant levels of pYSTAT1 or pYSTAT5. These data are now included in the revised version of our manuscript (**NEW Supplementary Fig. 6d**).

As demonstrated in **Fig. 2a and Supplementary Fig. 3e**, there are some differences in IL-2R α expression across the various cell lines; however, they all broadly express IL-2R α . There are no obvious differences seen in the killing efficiency following ADCT-301 treatment (see also **Fig. 4c**).

14. Would ADCT-301 also affect normal T cells, since these potentially also express ILR2A?

ADCT-301, binding human IL-2R α , and a close analogue, binding murine IL-2R α , have been reported to affect T_{regs}, an IL-2R α /CD25-expressing sub-population of T cells, in preclinical mouse models as well as in patients with advanced solid tumors (both in the circulation and in tumor tissue) [2,3] and with relapsed and refractory Hodgkin and non-Hodgkin lymphoma [4]. Interestingly, it has been shown in murine experiments and in patients with advanced solid tumors that CD4⁺CD25⁺ T_{reg} cells are transiently depleted with full recovery after 11 days after one dose of ADCT-301, leading to an anti-tumor immune response. In both the preclinical and clinical studies, the numbers of normal CD4 or CD8 T cells were not affected, and depletion of T_{regs} was accompanied by a concomitant increase in T_{eff}:T_{reg} ratio. We now refer to this issue and include references in the Discussion part of our manuscript. See references [4-6] below.

Reviewer #2, expert in lymphoma/clinical and genomics (Remarks to the Author):

1) Major claims of the paper

This is an interesting paper showing that BATF3 and IL2R chains are probably key regulators for the oncogenic activity in ALCL. It also shows that targeted inhibition of IL2Ra with an antibody-drug conjugate, effectively kills ALCL cells in vitro and in vivo making it a promising strategy in (relapsed/refractory) ALCL.

We very much appreciate that the reviewer calls our approach a “promising strategy”.

2) Are the findings novel and will they be of interest in the community and in the field? The biological findings described in this study are novel and will, if confirmed, have the potential to advance the field in terms of both biological insight and new therapeutic options for patients with ALCL. However, in contrast with some of the introductory and final statements in the manuscript, ALCL histology, in particular ALK⁺ ALCL, primary cutaneous (ALK⁻)ALCL and BIA-ALCL, is recognized as being associated with a significantly better prognosis than other aggressive T-cell lymphomas. Recently, biological information has improved the prognostic granularity within the largest subset of ALCL, i.e. adult systemic ALCL. Therefore, it would be useful to have this information for the presented cohort in order to fully understand the clinical

potential of the reported findings in ALCL. The ongoing NCT02432235 trial is currently evaluating the targeting of IL2Ra in different types of lymphoma, including those of T-cell origin.

We thank the reviewer for these interesting comments (see also our detailed answers below). We also agree with the reviewer that our findings are novel and will advance the field in terms of both biological insight and new therapeutic options.

3) Reproducibility of the presented work? The methodology of the study is thoroughly explained and should allow other groups to try to reproduce results in independent cohorts

We thank the reviewer for his comments supporting the aptness of our methodology to clarify the questions asked.

0) Is further evidence required to strengthen the conclusions?

The authors state in different occasions that there is an unmet clinical need for patients with ALCL type lymphoma. While it may be applicable in a subset of systemic ALK-neg ALCL, this statement is (as mentioned under pt.2) not entirely correct for all cases of ALK-neg ALCL. Recent reports (Parilla-Catellar et al, Blood 2014; Pedersen MB et al, Blood 2017) have shown that additional biological features, such as the presence of rearrangements in the DUSP22 or the TP63 gene are associated (under current treatment approaches) with a favourable or unfavourable prognosis, respectively. To show unequivocal efficacy of ILRa inhibition in the setting of TP63 rearranged ALK-neg systemic ALCL, would undoubtedly strengthen the conclusions of this paper.

We agree with the reviewer that the statement of unmet medical need for ALCL patients may not be applicable to all ALCL patients. We have therefore modified the manuscript accordingly. However, even in the era of brentuximab, the prognosis of relapsed/refractory ALCL patients is poor, and treatment success of a proportion of ALCL, ALK⁻ is modest (Knörr *et al.* 2020 [7], Mussolin *et al.* 2020 [8], Parrilla *et al.* 2014[9]). We agree that it might be interesting to investigate IL-2R α inhibition in the context of TP63-rearranged systemic ALCL, ALK⁻, especially on this point, but unfortunately we are not aware of a satisfactory ALCL model that we could use for these experiments.

0) Any further question deemed relevant in this context?

(i) In the introductory sentence of the Abstract, the authors give an overview of ALCL subtypes. The primary cutaneous (consistently ALK-neg) ALCL is missing. It should be mentioned as an entity with good prognosis despite of ALK-negativity.

We have modified the abstract accordingly.

(ii) page 4, line 24: It would further strengthen the biologic observation of this study, if the authors could show that also cases of primary cutaneous ALCL display a strong expression of IL2R chains and/or BATF3 similarly to the other ALCL subgroups. In fact, suppl.Table 3 shows that the primary tissue biopsy used for TMA construction in this study was 'SKIN' in 6 (10%) of the tabulated 60 ALK-neg ALCL pts. Could it be possible that some of these cases were primary cutaneous ALCLs?

To address this important point of the reviewer and to further increase the depth of our study, we have now assessed the BATF3, IL-2R α , IL-2R13 and IL-2R γ status using IHC in FFPE tissue samples of 24 primary cutaneous ALCL cases. Of these cases, 14/22 express BATF3 (64%), 20/23 express IL-2R α (87%), 20/22 express IL-2R13 (91%) and finally 16/16 (100%) express IL-2R γ . We also find a positive correlation of BATF3 expression with IL-2R α expression in primary cutaneous ALCL ($r = 0.6047$; $P = 0.004$), but BATF3 expression is somewhat lower compared to systemic ALCL subtypes. However, these data indicate widespread and concomitant expression of BATF3, IL-2R α , IL-2R13 and IL-2R γ in primary cutaneous ALCL. These data are now included in the NEW version of the manuscript (**NEW Fig. 1h, NEW**

Supplementary Fig. 2f, NEW Supplementary Fig. 3b and NEW Supplementary Table 4), and the manuscript has been modified accordingly.

As the reviewer has rightly pointed out, among the ALCL, ALK⁻ cases referred to in **Supplementary Table 3**, two have "skin" indicated as tissue origin (the other 4 cases refer not to be ALCL, ALK⁻). Both cases are positive for BATF3, IL-2R α , IL-2R β and IL-2R γ . Whether these cases were primary or secondary ALCL is, due the lack of clinical data, not to judge. However, with our newly stained cases of primary cutaneous ALCL, we hope that the reviewer agrees with our view that the question of expression of the respective genes in primary cutaneous ALCL is clarified.

(iii) page 6, line 14: The pts from the NHL-BFM and the ALCL99 studies were treated in trials run 20 and 10 yrs ago, respectively, i.e. prior to the advent of brentuximab vedotin. The outcomes for both ALK⁺ ALCL subsets (IL2Ra low and IL2R high) shown in Fig.4a are high (best group 100% OS at 5 yrs; worst group 80% OS at 5 yrs). Not exactly an 'unmet clinical need', at least in the first-line setting. Moreover, one could assume that the addition of brentuximab vedotin to the treatment schedule could improve outcome also for pts with high IL2Ra levels to values above 80% 5-yr OS. This should be mentioned/discussed.

First-line therapy for pediatric ALCL, ALK⁺ has not changed during the last 30 years. The current worldwide standard of care is six courses ALCL99-type chemotherapy without BV or an ALK-inhibitor, e.g. crizotinib. The first phase II study combining ALCL99 with either BV or crizotinib (ANHL12P1) has fully recruited. However, the results have not been published yet and the primary endpoint is toxicity. There are currently no data available suggesting that the addition of BV decreases the relapse rate.

We respectfully disagree with the reviewer that there is no medical need in first-line therapy for ALCL, ALK⁺ in children. The relapse rate constantly reaches 30% (e.g. Mussolin *et al.*, Cancers 2020 [7]) with any chemotherapy regimen, and relapse therapy includes an allogeneic blood stem cell transplantation (SCT) after total-body-irradiation-based conditioning for most relapsed patients (e.g. Knörr *et al.*, JCO 2020 [8]). This implies that all children bear the risks of late effects of systemic polychemotherapy, 7-10% die of disease, and 25% have the substantial late effects risk after allogeneic SCT (in addition to a TRM of 5-10% after SCT). These papers are now cited and discussed in the introduction section of the new version of the manuscript.

(iv) The patient cohort of adult ALK-neg ALCL pts shown in Fig.4a is very small, i.e. 15 pts subdivided in 2 groups: 5 pts with low IL2Ra expression (better outcome) and 10 with high IL2Ra expression (worse outcome). A larger cohort size would allow for more robust conclusions.

What was the DUSP22 and TP63 rearrangement status of these pts? What was the individual treatment background of these patients? Were there any cases with primary skin involvement in the IL2Ra 'high' group?

We agree with the reviewer that the cohort of adult ALCL, ALK⁻ in the first version of our manuscript was small. Unfortunately, ALCL, ALK⁻ patient cohorts with annotated clinical data are rare, and even rarer are cohorts with clinical data in combination with available material for IHC analyses. In the European countries of the main co-authors (Germany and Austria) no such cohorts are available. However, in an effort to extend our cohort, we contacted the large lymphoma study groups across Europe.

At the end, we succeeded to obtain clinical data and lymphoma tissue sections from a total of 29 additional ALCL, ALK⁻ patients from the French / Swiss lymphoma study groups (David Sibon / Philippe Gaulard / Laurence de Leval). By combining these three cohorts, we now

provide in the revised version of our manuscript data for an extended ALCL, ALK⁻ cohort with a total of 44 patients. This is to our knowledge one of the largest ALCL, ALK⁻ cohorts available so far. The results for EFS and OS are now included in **NEW Fig. 4b** and **NEW Supplementary Fig. 6a, NEW Supplementary Table 6**

We performed two analyses:

1) Estimates for EFS and OS at 3 years for all 44 patients by IL-2R α staining intensity (SI = 0, 1, 2, 3) showed a possible difference in EFS, not reaching significance (EFS 32% [95%-CI, 14–51%] for strong IL-2R α positivity (SI = 3) vs. 50% [95%-CI, 28–68%], $P = 0.11$).

First-line treatment for the patients was comparable in terms of induction chemotherapy, but differed in consolidation strategy, as 10 of 44 patients received an autologous or allogeneic SCT first-line.

2) Therefore, we performed a subgroup analysis for patients not receiving SCT as first-line consolidation. In this analysis, EFS was significantly lower in patients with tumors showing strong IL-2R α staining (SI = 3) (24% [95%-CI: 7–45%] vs. 59% [95%-CI: 33–78%, $P = 0.025$]. No significant difference was observed in OS ($P = 0.056$). Low patient numbers and heterogeneity between the patients and treatments preclude definitive conclusions, but these clinical data support our initial statement that high-level IL-2R α -expression is associated with a poor prognosis. As risk factor, IL-2R α expression might be outcompeted by more intense treatment regimen with SCT. Larger prospective patient cohorts, which are currently not available, could be used to validate our findings. We modified the manuscript accordingly (**NEW Fig. 4b** and **NEW Supplementary Fig 6a**).

Regarding *DUSP22* and *TP63*, we do not have the *DUSP22* and *TP63* status of the patients used for Kaplan-Meier in **NEW Fig. 4b**. All necessary clinical information including the individual treatment background of the patients are now included in **NEW Supplementary Table 6**. All cases had systemic ALCL, ALK⁻.

(v)page 7, line 32 one line from the bottom): Alk-inhibitors (e.g. crizotinib) should be mentioned in the context of ALK⁺ ALCL (e.g.Mossè YP, JCO 2017).

We agree with the reviewer and now mention ALK-inhibitors with the respective citations in the context of ALCL, ALK⁺ in the revised version of the manuscript.

() page 8, line 1: The ECHELON-2 study mentioned did not (was not powered to) demonstrate superiority in other entities than ALCL. The effect of brentuximab vedotin in the trial was most pronounced in ALK⁺ than ALK-neg ALCL. Also, a subset analysis performed on the ECHELON-2 dataset (Savage K et al, ASH 2019, oral presentation) showed, in ALK-ALCL pts, a possible benefit of autologous transplant consolidation in BV-treated ALCL pts. No information on *DUSP22* and *TP63* status was available from the ECHELON-2 data set.

We thank the reviewer for this comment and refer in the revised version of our manuscript in the respective sentence now only to ALCL.

(i) page 8, line 9 (last sentence): This point would be significantly strengthened, if the authors could demonstrate efficacy in *TP63*-rearranged ALK-neg ALCL

We agree that it would be interesting to see the activity of this antibody-drug conjugate (ADC) in ALCL, ALK⁻ cells with *TP63*-rearrangement. Unfortunately, there is no model system to test this as the only ALCL cell line with *TP63*-rearrangement is DL40. Unfortunately, it is IL-2R α -negative; therefore, it would not be recognized by our ADC.

Reviewer #3, expert in lymphoma/clinical and genomics (Remarks to the Author):

The authors use analysis of superenhancers in cell lines to identify a regulatory module of BATF3 with IL2 receptor genes that facilitates IL2/IL15-mediated activation of STATs, enhances cell growth, and can be targeted with an anti-IL2RA ADC. The studies are well conducted and the overall findings convincing.

1. The authors should temper their statement that the “prognosis [of ALCL] is unsatisfactory” with data on the often good prognosis in ALK⁺ ALCL, some subsets of ALK⁻ ALCL (especially in BV era), and cutaneous ALCL, which is not mentioned at all and should be addressed. Also, they need to acknowledge that Mac-1, 1 of 2 lines on which the original SE discovery was performed, is not a true ALCL cell line, having been derived from Sezary-like cells from the peripheral blood of a patient with lymphomatoid papulosis and cutaneous T-cell lymphoma who only later progressed to large cell lymphoma (PMID: 1532439). It is also somewhat overstated that the BATF3/IL2R module is a “new vulnerability” in ALCL since only IL2R (and not BATF3) is proposed as a direct target, and the expression of IL2R in ALCL and its potential as a target have been known for some time.

We agree with the reviewer regarding the often-good prognosis in particular of ALCL, ALK⁺ patients, some subsets of ALCL, ALK⁻ patients and primary cutaneous ALCL (which we now mention in the revised version of our manuscript). However, in our view, there is a clear medical need to improve treatment success in ALCL, ALK⁻ in the first-line setting as well as ALCL in general in the relapse setting. Furthermore, there is also a medical need in first-line therapy for ALCL, ALK⁺ in children. The relapse rate constantly reaches 30% (e.g. Mussolin *et al.*, Cancers, 2020[7]) with any chemotherapy regimen, and relapse therapy includes an allogeneic SCT after total-body-irradiation-based conditioning for most relapsed patients (e.g. Knörr *et al.*, JCO, 2020 [8]). This implies that all children bear the risks of late effects of systemic polychemotherapy, 7-10% die of disease, and 25% have substantial late effect risk after allogeneic SCT (in addition to a TRM of 5-10% after SCT). However, we modified the manuscript accordingly and tuned down our overall statement that “prognosis [of ALCL] is unsatisfactory”.

Regarding the origin of Mac-1 cells, we thank the reviewer for this comment and we have now described the Mac-1 cell line in more detail in order to avoid misunderstandings. Also, to further increase the depth of our study we have now performed and included SE profiling of 6 additional ALCL cell lines (4 ALCL, ALK⁺ cell lines: SU-DHL-1, SR-786, JB6, SUP-M2; 2 ALCL, ALK⁻ cell lines: Mac-2A and FE-PD), 2 ALCL patient samples, and analyzed publicly available datasets of CD3, CD4 and CD8 T cells from healthy donors (**NEW Fig. 1d** and **NEW Supplementary Fig. 1a, b, c**). These analyses all confirmed our previous results and demonstrate the presence of SEs at *BATF3*, *IL-2RA* and *IL-2RB* in the analyzed ALCL samples, whereas in the normal T cells no SE formation was observed.

Also, to clarify the relevance of our work for primary cutaneous ALCL, we have now analyzed 24 primary cutaneous ALCL cases for the expression of BATF3, IL-2R α , IL-2R β and IL-2R γ using IHC analyses. These data are now included in our manuscript and confirm our findings also applied for primary cutaneous ALCL (**NEW Fig. 1h**, **NEW Supplementary Fig. 3b** and **NEW Supplementary Table 4**). In addition, we have replaced “new vulnerabilities” with

“vulnerabilities” and thus, deleted the word “new” in the respective phrases. We have also deleted the word “new” in the title of the revised manuscript.

2. The hypothesis “that IL2 might originate from the tumor environment” is not adequately tested by the methodology and the image in Fig. 3a is insufficient to justify this. This should either be removed or replaced by quantitative and properly controlled experiments to test this hypothesis.

We have now removed Fig. 3a and moved it to **NEW Supplementary Fig. 4a** and show IHC picture of an additional ALCL patient to address concerns of reviewer. Moreover, we have analyzed 33 additional ALCL patients for IL-2 expression and found them all negative (**NEW Supplementary Table 3** and **NEW Supplementary Fig. 4a**), supporting that IL-2 is not expressed by primary ALCL tumor cells. We have also toned down the claim that IL-2 comes from the tumor microenvironment in the results section of the current version of the manuscript. Elevated IL-2 levels were previously found in the serum of a subgroup of ALCL patients but none was detected upon remission or in B-NHL controls (Knorr *et al.* 2018 [10]).

References:

1. Schleussner, N., et al., The AP-1-BATF and -BATF3 module is essential for growth, survival and TH17/ILC3 skewing of anaplastic large cell lymphoma. *Leukemia* 32, 2018
2. Watanabe, M., *et al.*, AP-1 mediated relief of repressive activity of the CD30 promoter microsatellite in Hodgkin and Reed-Sternberg cells. *The American journal of pathology*, 163, 2003
3. Watanabe, M., *et al.*, JunB induced by constitutive CD30-extracellular signal-regulated kinase 1/2 mitogen-activated protein kinase signaling activates the CD30 promoter in anaplastic large cell lymphoma and reed-sternberg cells of Hodgkin lymphoma, *Cancer research* 65, 2005
4. Zammarchi, F., et al., *CD25-targeted antibody-drug conjugate depletes regulatory T cells and eliminates established syngeneic tumors via antitumor immunity.* *J Immunother Cancer*, 8(2), 2020..
5. Puzanov, I., et al., *First-in-Human Study of Camidanlumab Tesirine (ADCT-301, Cami), an anti-CD25 Targeted Therapy in Patients with Advanced Solid Tumors: Pharmacokinetics (PK) and Biomarker Evaluation.* *Annals of Oncology* 32, 2020
6. Boni, J., et al., *Pharmacokinetic and Pharmacodynamic Correlates from the Phase 1 Study of Camidanlumab Tesirine (Cami) in Patients with Relapsed or Refractory Hodgkin Lymphoma and Non-Hodgkin Lymphoma* *Blood*, 126 (Supplement 1): p. 35-36, 2020.
7. Mussolin, L., et al., Prognostic Factors in Childhood Anaplastic Large Cell Lymphoma: Long Term Results of the International ALCL99 Trial, *Cancers (Basel)*, 2020
8. Knörr, F., et al., Stem Cell Transplantation and Vinblastine Monotherapy for Relapsed Pediatric Anaplastic Large Cell Lymphoma: Results of the International, Prospective ALCL-Relapse Trial, *Journal of Clinical Oncology*, 2020
9. Parrilla Castellar, E.R., et al. ALK-negative anaplastic large cell lymphoma is a genetically heterogeneous disease with widely disparate clinical outcomes. *Blood* 124, 1473-1480, 2014
0. **Knorr, F., et al. Blood cytokine concentrations in pediatric patients with anaplastic lymphoma kinase-positive anaplastic large cell lymphoma. *Haematologica* 103, 477-485, 2018**

Reviewers' Comments:

Reviewer #1:

Remarks to the Author:

All my concerns have been addressed.

Reviewer #2:

Remarks to the Author:

The authors should be acknowledged for having provided extensive and valuable additional information based on most reviewer comments.

However, considering the fact that this paper exclusively focuses on anaplastic large cell lymphomas (ALCL) and that systemic ALK-neg ALCL (mostly occurring in adults) represent the majority of ALCL cases in the general population, it is remarkable that the DUSP22 and TP63 rearrangement status of systemic ALK-neg ALCL cases has not been established (probes are available and can be analyzed on paraffin tissue) or even discussed in any of the manuscript sections. These molecular features have prognostic relevance in systemic ALK-neg ALCL and have been added into the ALCL chapter of the 2016/2017 revision of the WHO classification of Lymphoid Neoplasms.

According to the data published so far, approximately 20% and 5% of systemic ALK-neg ALCL are DUSP22 or TP63 rearranged, respectively. Their potential occurrence in the IL2Ra-low and IL2Ra-high subcohorts shown in Fig.4b and suppl.Fig.6a should be established and adjusted for, or, if not feasible, at least discussed as a potential prognostic confounder.

Reviewer #3:

Remarks to the Author:

Thank you for the detailed responses. My comments have been addressed adequately.

Reviewer #1 (Remarks to the Author):

All my concerns have been addressed.

We are happy that we were able to address all the 14 questions of reviewer 1 and appreciate that he accepts the manuscript in its present form.

Reviewer #2 (Remarks to the Author):

The authors should be acknowledged for having provided extensive and valuable additional information based on most reviewer comments.

However, considering the fact that this paper exclusively focuses on anaplastic large cell lymphomas (ALCL) and that systemic ALK-neg ALCL (mostly occurring in adults) represent the majority of ALCL cases in the general population, it is remarkable that the *DUSP22* and *TP63* rearrangement status of systemic ALK-neg ALCL cases has not been established (probes are available and can be analyzed on paraffin tissue) or even discussed in any of the manuscript sections. These molecular features have prognostic relevance in systemic ALK-neg ALCL and have been added into the ALCL chapter of the 2016/2017 revision of the WHO classification of Lymphoid Neoplasms.

According to the data published so far, approximately 20% and 5% of systemic ALK-neg ALCL are *DUSP22* or *TP63* rearranged, respectively. Their potential occurrence in the IL2R α -low and IL2R α -high subcohorts shown in Fig.4b and suppl.Fig.6a should be established and adjusted for, or, if not feasible, at least discussed as a potential prognostic confounder.

To address the final concerns of reviewer 2, we have now inserted the following sentence in the Discussion section of our manuscript.

"Moreover, it is unlikely that *DUSP22* and *TP63* rearrangements, having been described in ALCL, ALK⁻ (Feldman et al., 2009; Vasmatazis et al., 2012), are confounding factors in our survival analysis based on the following findings. In the first clinical report, the 5-year OS of 21 patients with *DUSP22* rearrangement was 90% (Parrilla Castellar et al., 2014). However, in a later work from the British Columbia Cancer Lymphoid Cancer database, the 5-year OS of 12 patients with *DUSP22* rearrangement was 40% (Hapgood et al., 2019). Thus, the prognostic impact of *DUSP22* rearrangement is currently unclear, and it may be more related to prognostic factors such as IPI. In addition, in pcALCL, no impact of *DUSP22* rearrangement (present in about 30% of cases) on prognosis was observed (Fauconneau et al., 2015). Despite that *TP63* rearrangement has a confirmed poor prognosis in ALCL, ALK⁻, it is indeed very rare."

Reviewer #3 (Remarks to the Author):

Thank you for the detailed responses. My comments have been addressed adequately.

We thank reviewer 3 for his positive view of our manuscript.

Best regards and thank you for your support,

Olaf Merkel on behalf of coauthors

IRF4 and DUSP22 are neighbouring genes located on locus 6p25. Feldman et al.²² showed that IRF4 locus rearrangement was a recurrent cytogenetic abnormality in 57% of the 12 cALCL studied. This was subsequently confirmed with 20–75% of cALCL displaying IRF4 locus gene rearrangement.^{23,24,48} Some TMF tested in those studies also displayed 6p25 locus rearrangement but at a lower rate.²³ Wada et al.²⁴ branded the IRF4 translocation specific for cALCL with a positive predictive value close to 100%. The results of a previous report of our group²³ prompt us to draw more cautious conclusions. Indeed, IRF4 translocation was more frequent in cALCL but was also found in some cases of TMF. As previously reported in several small series,^{23,24} this rearrangement had no prognostic impact among cALCL. It has been shown recently that translocation on this locus often does not alter IRF4 gene expression but rather that of one of the neighbouring genes, DUSP22.²⁵ Thus, we tried to find whether DUSP22 protein loss, in IHC, was correlated with the 6p25 locus translocation. This procedure has the potential advantage of being easier than FISH to carry out for screening. However, given the low specificity of the available antibody providing signals on paraffin sections, such analysis could not elicit significant differences in DUSP22 immunoreactivity between cases with or without 6p25 rearrangement.

Feldman, A.L., Law, M., Remstein, E.D., Macon, W.R., Erickson, L.A., Grogg, K.L., Kurtin, P.J., and Dogan, A. (2009). Recurrent translocations involving the IRF4 oncogene locus in peripheral T-cell lymphomas. *Leukemia* 23, 574-580.

Vasmatazis, G., Johnson, S.H., Knudson, R.A., Ketterling, R.P., Braggio, E., Fonseca, R., Viswanatha, D.S., Law, M.E., Kip, N.S., Ozsan, N., Grebe, S.K., Frederick, L.A., Eckloff, B.W., Thompson, E.A., Kadin, M.E., Milosevic, D., Porcher, J.C., Asmann, Y.W., Smith, D.I., Kovtun, I.V., Ansell, S.M., Dogan, A., and Feldman, A.L.

(2012). Genome-wide analysis reveals recurrent structural abnormalities of TP63 and other p53-related genes in peripheral T-cell lymphomas. *Blood* 120, 2280-2289.

Parrilla Castellar, E.R., Jaffe, E.S., Said, J.W., Swerdlow, S.H., Ketterling, R.P., Knudson, R.A., Sidhu, J.S., Hsi, E.D., Karikehalli, S., Jiang, L., Vasmatazis, G., Gibson, S.E., Ondrejka, S., Nicolae, A., Grogg, K.L., Allmer, C., Ristow, K.M., Wilson, W.H., Macon, W.R., Law, M.E., Cerhan, J.R., Habermann, T.M., Ansell, S.M., Dogan, A., Maurer, M.J., and Feldman, A.L. (2014). ALK-negative anaplastic large cell lymphoma is a genetically heterogeneous disease with widely disparate clinical outcomes. *Blood* 124, 1473-1480.

Haggood, G., Ben-Neriah, S., Mottok, A., Lee, D.G., Robert, K., Villa, D., Sehn, L.H., Connors, J.M., Gascoyne, R.D., Feldman, A.L., Farinha, P., Steidl, C., Scott, D.W., Slack, G.W., and Savage, K.J. (2019). Identification of high-risk DUSP22-rearranged ALK-negative anaplastic large cell lymphoma. *British journal of haematology* 186, e28-e31.

Fauconneau, A., Pham-Ledard, A., Cappellen, D., Frison, E., Prochazkova-Carlotti, M., Parrens, M., Dalle, S., Joly, P., Viraben, R., Franck, F., Ingen-Housz-Oro, S., Giaccherio, D., Jullie, M.L., Vergier, B., Merlio, J.P., and Beylot-Barry, M. (2015). Assessment of diagnostic criteria between primary cutaneous anaplastic large-cell lymphoma and CD30-rich transformed mycosis fungoides; a study of 66 cases. *Br J Dermatol* 172, 1547-1554.